# ReAgent-V: A Reward-Driven Multi-Agent Framework for Video Understanding

**Yiyang Zhou**[1*]   **Yangfan He**[1*]   **Yaofeng Su**[1]   **Siwei Han**[1]   **Joel Jang**[2]
**Gedas Bertasius**[1]   **Mohit Bansal**[1]   **Huaxiu Yao**[1]
[1]UNC-Chapel Hill, [2]University of Washington
yiyangai@cs.unc.edu, huaxiu@cs.unc.edu

## Abstract

Video understanding is fundamental to tasks such as action recognition, video reasoning, and robotic control. Early video understanding methods based on large vision-language models (LVLMs) typically adopt a single-pass reasoning paradigm without dynamic feedback, limiting the model's capacity to self-correct and adapt in complex scenarios. Recent efforts have attempted to address this limitation by incorporating reward models and reinforcement learning to enhance reasoning, or by employing tool-agent frameworks. However, these approaches face several challenges, including high annotation costs, reward signals that fail to capture real-time reasoning states, and low inference efficiency. To overcome these issues, we propose ReAgent-V, a novel agentic video understanding framework that integrates efficient frame selection with real-time reward generation during inference. These reward signals not only guide iterative answer refinement through a multi-perspective reflection mechanism—adjusting predictions from conservative, neutral, and aggressive viewpoints—but also enable automatic filtering of high-quality data for supervised fine-tuning (SFT), direct preference optimization (DPO), and group relative policy optimization (GRPO). ReAgent-V is lightweight, modular, and extensible, supporting flexible tool integration tailored to diverse tasks. Extensive experiments on 12 datasets across three core applications—video understanding, video reasoning enhancement, and vision-language-action model alignment—demonstrate significant gains in generalization and reasoning, with improvements of up to 6.9%, 2.1%, and 9.8%, respectively, highlighting the effectiveness and versatility of the proposed framework. Our code are available at https://github.com/aiming-lab/ReAgent-V.

## 1   Introduction

Video understanding, as one of the core tasks in computer vision, is widely applied in scenarios such as action recognition, video reasoning, and robotic control. Although significant progress has been made in recent years through the application of large vision-language models (LVLMs) and multimodal transformers, most existing methods adopt a static reasoning paradigm—that is, the model generates predictions directly from the input prompt and video in a single pass, without reflection or reward-based feedback to guide the reasoning process [10, 24, 4, 56, 6]. This design limits the model's adaptability in complex scenarios that demand iterative reasoning or task-specific feedback. For example, in tasks involving multi-step reasoning or reward-based evaluation, the model must reason step by step, verify intermediate results through reflection, and potentially use reward signals to assess the accuracy of each step. Without such mechanisms, static reasoning hampers the model's ability to self-correct and optimize its reasoning trajectory.

---

[*]Equal contribution.

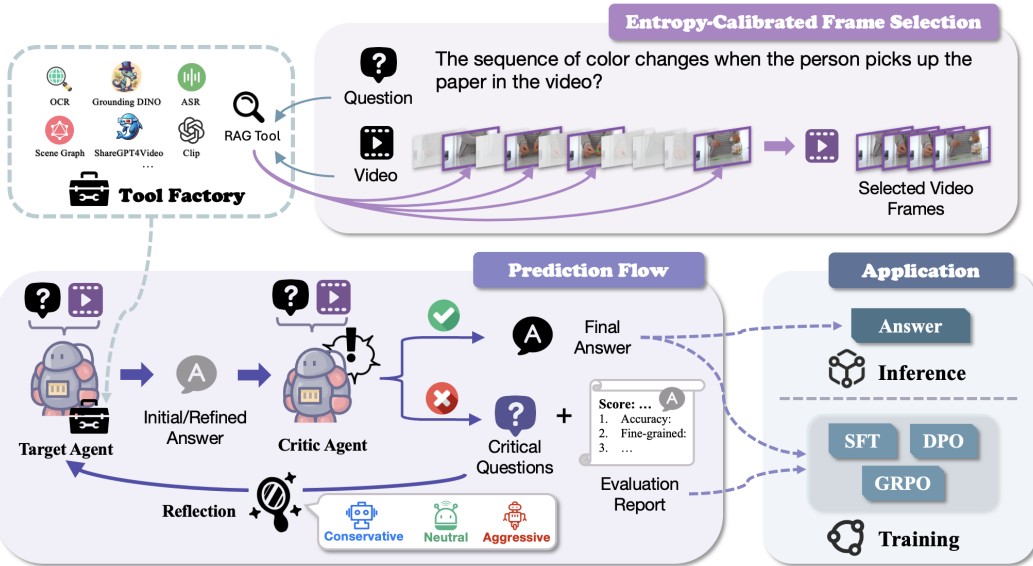

Figure 1: Overview of the ReAgent-V framework: The system first selects relevant video frames based on the input question through the entropy-calibrated frame selection module and invokes various tools from the tool factory to assist in reasoning. The target agent generates an initial answer using the selected tools and input context, which is then critically evaluated by the critic agent through questioning and scoring, ultimately producing a comprehensive feedback report. The target agent can then revise its answer from three perspectives—conservative, neutral, and aggressive—based on the report and updated context. In addition to generating standard reasoning outputs, high-scoring data identified during the reflection process based on the feedback report is stored for training algorithms such as SFT, DPO, and GRPO, thereby further enhancing model performance.

To improve video reasoning performance, previous research has mainly relied on two strategies: expanding high-quality annotated datasets for supervised fine-tuning (SFT) and introducing external reward models or reward templates to enhance the base model through reinforcement learning [10, 33, 54, 20]. While both approaches offer certain benefits, they also face notable limitations. On the one hand, obtaining high-quality annotations is often prohibitively expensive, limiting the scalability of methods that depend on increasing the volume of labeled data. On the other hand, using pretrained reward models for offline evaluation fails to capture the model's real-time reasoning state during inference, which can lead to suboptimal strategies or even reinforce flawed reasoning patterns. Although reward templates provide some form of real-time feedback, they are typically static and lack the adaptability needed to handle complex tasks or respond dynamically to changes during the reasoning process. For example, reward templates are typically predefined and are limited to providing feedback on multiple-choice answers or responses that follow specific templates, which makes it challenging to evaluate the reasoning process underlying a model's answer. More recently, approaches based on multi-model debate or tool-agent frameworks have been proposed to enhance reasoning capabilities [40, 9, 52]. However, these methods are often time-consuming and inefficient, with limited functionality, and they lack the ability to provide reward signals during inference.

To address these limitations, we propose an innovative agent-based video understanding framework, ReAgent-V, as shown in Figure 1. The core idea of our framework not only utilizes a dedicated frame selection module for efficient video understanding during inference, but also generates real-time reward signals. These signals help the model refine its answers, and at the same time, they can be used to filter high-quality data for SFT, DPO [33], and GRPO [34], enabling continuous optimization through learning during inference. To further enhance this process, ReAgent-V incorporates an intelligent reflection mechanism that prompts the target agent to revise its answers from conservative, neutral, and aggressive perspectives, where conservative only adjusts the final answer, neutral updates entities in the scene based on the input context, and aggressive modifies both the reasoning steps and the involved entities. This multi-perspective evaluation helps mitigate biases caused by model overconfidence and leads to more robust and diverse answer refinement. Additionally, ReAgent-V is designed with simplicity and extensibility in mind. It provides a modular architecture that allows users to customize the base models of both the target agent and the critic agent, as well as the task

templates. ReAgent-V supports a variety of applications, such as enhanced video understanding and video-based model rewarding. The former aims to improve the video reasoning capability of the existing base model, while the latter leverages reward feedback during the reasoning process to collect high-quality data, which is then used for training to further enhance the model's performance.

In conclusion, the key contribution of this work is ReAgent-V, the unified video understanding agentic framework capable of providing real-time rewarding during inference. Through extensive experiments on 12 datasets or tasks across three major applications—video understanding, video LLM reasoning, and vision-language action model alignment. ReAgent-V achieves performance improvements up to 6.9%, 2.1%, and 9.8%, respectively. Additionally, our study highlights the contribution of the frame selection mechanism in ReAgent-V and the intelligent reflection system, both of which are essential for improving the reasoning efficiency of the agent framework, reducing overconfidence bias, and enhancing the accuracy of reward feedback.

## 2 ReAgent-V

To improve video understanding and support dynamic reward feedback throughout the inference process, we propose a lightweight, general, and extensible agent framework, ReAgent-V. As illustrated in Figure 1, ReAgent-V enhances the reasoning process of the target agent through three key stages: entropy-calibrated frame selection, tool-augmented reasoning, and intelligent reflection with dynamically generated reward signals. First, the framework selects task-relevant video frames using an entropy-guided sampling strategy (§2.1), reducing redundancy and constructing a concise semantic context. Next, the target model performs initial reasoning by invoking appropriate external tools (§2.2). Finally, based on the reward signals provided by the critic agent, the target agent revises its initial answer from multiple perspectives—conservative, neutral, and aggressive—thereby improving the model's adaptability and robustness (§2.3). Detailed descriptions of each component are provided in the following sections, and the overall process is outlined in Algorithm 1.

### 2.1 Entropy-Calibrated Frame Selection

In this subsection, we introduce the keyframe selection strategy adopted in ReAgent-V. Common methods typically rely on generating captions for each frame or directly querying LVLMs such as GPT-4o to select keyframes that are most relevant to the prompt [49, 40]. However, such approaches are inefficient and slow in practice. Others focuses on CLIP-based selection [30, 48], where frames are chosen based on the CLIP score between each frame and the prompt. While efficient, this approach often fails on questions like "Does

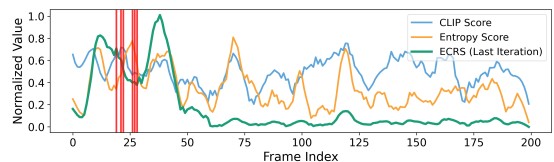

Figure 2: Frame selection analysis (VideoMME VideoID: 24i4ncHuf6A, QuestionID:005-2) shows entropy, CLIP score, and ECRS across frames; red lines highlight the most relevant frames.

the person wash their hands before or after placing the cup?", where many frames contain a person, resulting in high CLIP scores even for irrelevant moments. This leads to the inclusion of redundant frames that are semantically unrelated to the question. We conducted a statistical analysis of this phenomenon, with detailed experiments reported in Appendix D. One example is shown in Figure 2, where the frames relevant to the question are manually annotated (as indicated by the red lines) after watching the video. We can observe that selecting frames solely based on the CLIP score tends to result in redundant frames.

To address this limitation, we propose a metric called the Entropy-Calibrated Relevance Score (ECRS), which jointly considers the semantic relevance of each frame to the query and the information entropy of each frame—i.e., the amount of information it contains—as illustrated by the orange line in Figure 2. Specifically, given a video $V = \{f_1, f_2, \ldots, f_N\}$ consisting of $N$ frames and a query $Q$, we first extract feature embeddings for each frame $f_i$ and the query $Q$ using a shared CLIP encoder:

$$\mathbf{e}_i = T_{\text{CLIP}}(f_i), \quad \mathbf{q} = T_{\text{CLIP}}(Q), \tag{1}$$

where $\mathbf{e}_i$ and $\mathbf{q}$ represent the embedding vectors of the $i$-th frame and the query, respectively. The semantic similarity between each frame and the query is then computed by the cosine similarity $s_i = \frac{\mathbf{e}_i^\top \mathbf{q}}{\|\mathbf{e}_i\|\|\mathbf{q}\|}$, quantifies how well each frame $f_i$ aligns with the query $Q$. To incorporate frame-level

**Algorithm 1** ReAgent-V Video Understanding Pipeline

---

**Require:** Video $V$; Query $Q$; Toolset $\mathcal{T}$; Prompt for critics agent $Q_c$; Reflection prompt for target agent $\{t_c, t_n, t_a\}$
**Ensure:** Final answer $A_{\text{final}}$; Evaluation report $E$
1: **Stage 1: Frame Selection (§2.1)**
2: Compute ECRS using Eqn. (3).
3: **while** there are frames meeting the threshold ECRS $> k \cdot \alpha^m \cdot \tau$ **do**:
4:     Save the current round's selected frames $F_m$.
5:     Recompute ECRS in set $F_m$.
6: Obtain the selected frame set $F$
7: **Stage 2: Tool-Augmented Reasoning (§2.2)**
8: Target agent select tools $\mathcal{T}' \subseteq \mathcal{T}$
9: Obtain tool information $R$ via $T_i(Q, F)$, where $T_i$ in $\mathcal{T}'$
10: Provide the target agent with $F$, $Q$, and $R$ to obtain the initial answer $A_0$.
11: **Stage 3: Evaluation and Reflection (§2.3)**
12: **if** Critic agent rejects $A_0$ **then**
13:     Generate questions $\{q_i\}_{i=1}^N$.
14:     Select new tools from $\mathcal{T}$, update $R_{update} \leftarrow R$, and generate evaluation report $E$.
15:     Target agent generate reflected answers and confidence score $(A^{(t)}, p^{(t)})$ for $t \in \{t_c, t_n, t_a\}$ from $R_{update}$, $Q$, $F$ and $E$
16:     **if** $\min(p^{(c)}, p^{(n)}, p^{(a)}) > 0.6$:
17:         $A_{\text{final}} \leftarrow$ merged reliable parts of $A^{(t)}$ based on $E$ and $Q$, where $t \in \{t_c, t_n, t_a\}$
18:     **else**: $A_{\text{final}} \leftarrow A^{(t)}$, where $t = \arg\max_{t \in \{t_c, t_n, t_a\}} p^{(t)}$
19: **end if**
20: **return** $A_{\text{final}}, E$

---

information entropy, we denote the entropy of the $i$-th frame as $H_i$. For color images, we compute the entropy separately for each RGB channel and take the average across channels:

$$H_i = \frac{1}{3} \sum_{c \in \{R,G,B\}} \left( -\sum_{j=0}^{255} p_{j,c}^{(i)} \log_2 p_{j,c}^{(i)} \right), \tag{2}$$

where $p_{j,c}^{(i)}$ represents the probability of pixel value $j$ in channel $c$ of the $i$-th frame. Finally, the ECRS for each frame is defined as:

$$\text{ECRS}_i = \frac{s_i \cdot H_i}{\sum_{k=1}^N H_k}, \tag{3}$$

where $N$ is the total number of frames selected in the current frame set. This score jointly considers the similarity between each frame and $Q$, as well as the amount of information contained in each frame. After obtaining $\text{ECRS}_i$, frame selection is performed iteratively. We set a base threshold $\tau$ and an iteration index $m$ starting from 1, a frame $f_i$ is selected if:

$$\text{ECRS}_i > k \cdot \alpha^m \cdot \tau. \tag{4}$$

The term $k \cdot \alpha^m$ serves as a scaling factor for the selection threshold. The scaling factor in each iteration allows the threshold to increase exponentially, thereby ensuring that frames selected in subsequent iterations have higher ECRS. In each iteration $m$, a set of candidate frames $F_m = \{f_1, f_2, \ldots, f_c\}$ is selected, where $c$ denotes the number of frames chosen in the current round. For each frame in $F_m$, we recalculate its score $\text{ECRS}_i$ using Eqn. (3), and apply Eqn. (4) again to select new frames. This process continues until no additional frames meet the threshold, in which case we return the frame set selected from the previous iteration. If the number of frames is below 32, we select the non-repetitive frames with the highest $\text{ECRS}_i$ from the previous round's frame set to complete the selection. As shown in the lower part of Figure 2, our method consistently increases the ECRS of key regions after iteration while suppressing those of irrelevant areas, leading to more accurate frame selection.

## 2.2 Tool-Augmented Reasoning

After selecting the key frames, to enhance the reasoning capability of ReAgent-V in video understanding tasks, we design a tool-augmented iterative reasoning mechanism. This mechanism allows

target agent to dynamically select and apply tools based on task requirements, iteratively refining its understanding through multiple rounds of information exchange, resulting in higher accuracy and interpretability. Specifically, given the user query $Q$, the selected key frame set $F$, and a tool set $\mathcal{T}$, the target agent proactively selects a subset of tools $\mathcal{T}' \subseteq \mathcal{T}$ based on its reasoning needs, and applies them to the user query $Q$ and the selected key frame set $F$, for each tool $T_i$ in $\mathcal{T}'$, generating intermediate tool results $R_i = T_i(Q, F)$. These intermediate results, along with $Q$ and $F$, are fed into the target model, producing an initial answer $A_0$.

## 2.3 Evaluation and Multi-Perspective Reflection

After the target agent generates an initial answer $A_0$ based on the selected frames $F$, query $Q$, and tool outputs $R$, the critic agent evaluates the quality of $A_0$. If the answer is deemed unsatisfactory, the critic agent first generates a set of sub-questions $\{q_1, q_2, \ldots, q_l\}$ based on the context to help localize potential errors. Using these sub-questions, the critic agent supplements the original toolset $\mathcal{T}'$ with additional tools as needed and applies them to $F$ to obtain updated tool outputs $R_{\text{update}}$. The critic agent then uses this updated information to produce a process reward for the initial answer in the form of an overall evaluation report $E$, which includes both a scalar reward score and structured feedback on initial answer $A_0$ (see Appendix C for examples of evaluation reports).

Based on the evaluation report, the target agent performs reflective reasoning from three strategic perspectives: conservative ($t_c$), neutral ($t_n$), and aggressive ($t_a$). The conservative strategy focuses solely on adjusting the final answer, the neutral strategy updates the relevant entities in the scene (i.e., objects or elements related to the question) based on the given context, and the aggressive strategy revises both the reasoning steps and these entities. Prompt templates for each strategy are provided in Appendix B. Each strategy independently generates a revised answer $A^{(t)}$ and an associated confidence score $p^{(t)}$, where $t \in \{t_c, t_n, t_a\}$. The confidence score represents the model's estimated probability of the revised answer given the context:

$$p^{(t)} = P(A^{(t)} \mid F, Q, R_{\text{update}}, E). \tag{5}$$

Finally, if all confidence scores $p^{(t)}$ exceed a predefined threshold, the target agent aggregates the three revised answers $\{A^{(t_c)}, A^{(t_n)}, A^{(t_a)}\}$, extracts their common components, and removes inconsistencies to generate the final answer $A_{\text{final}}$. Otherwise, the system selects the answer with the highest confidence.

# 3 Experiments

In this section, we demonstrate the effectiveness of ReAgent-V across three distinct video-related applications by addressing the following key questions: (1) Can ReAgent-V enhance performance in video understanding? (2) Does the feedback reward mechanism in ReAgent-V effectively select data that improves the reasoning capabilities of video LLMs? (3) Can the preference data generated through ReAgent-V improve alignment in preference optimization? (4) Are the frame selection and reflection mechanisms in ReAgent-V beneficial?

## 3.1 Applications and Experimental Setups

This section outlines three applications and describes the experimental settings, evaluation benchmarks, and baseline models. The applications include video understanding, video LLM reasoning, and vision-language-action (VLA) model alignment. Further details are provided below.

**Video Understanding.** This application evaluates the effectiveness of ReAgent-V in enhancing base models for video understanding. We apply ReAgent-V to a range of base models with varying sizes and architectures, including LLaVA-Video-7B/72B [56] and the Qwen series [46, 38] (Qwen2.5-VL-7B/72B and Qwen2-VL-7B), and assess their performance across six widely used benchmarks: LongBench [39], NextQA [45], EgoSchema [31], LVBench [39], MLVU [59], and VideoMME [11]. To provide a comprehensive evaluation, we also include popular proprietary models [14, 37], open-source video-based LVLMs [4, 21, 55, 5, 41, 15], and general video agent frameworks [40]. For fair comparison, we re-evaluate all baselines (except proprietary models) using a similar number of video frames. During evaluation, both the target and critic models are set to the ReAgent-V-enhanced model itself. Detailed descriptions of the task prompt templates, the tool list, benchmark datasets,

Table 1: A comparison of ReAgent-V with state-of-the-art video-language models and general video agent frameworks across different benchmarks. The results of baselines are copied from [59, 9, 40, 2, 30]. "w/ sub" in VideoMME means that the model's input during evaluation includes the video's subtitle.

| Model | Frames | LongBench | NextQA | Egoschema subset | LVBench | MLVU | VideoMME[w/ sub.] | | | |
|---|---|---|---|---|---|---|---|---|---|---|
| | | | | | | | Short | Medium | Long | Overall |
| GPT-4o [14] | 384 | - | - | 72.2 | 34.7 | 64.6 | 80.0 | 70.3 | 65.3 | 71.9 |
| Gemini-1.5-Pro [37] | 0.5 fps | - | - | 71.1 | 33.1 | - | 81.7 | 74.3 | 67.4 | 75.0 |
| ShareGPT4Video-8B [4] | 16 | - | - | - | 39.7 | 46.4 | 48.2 | 36.3 | 35.0 | 39.9 |
| VideoChat2-7B [21] | 16 | - | - | 56.7 | 39.3 | 47.9 | 48.3 | 37.0 | 33.2 | 39.5 |
| LLaVA-NeXT-Video-7B [55] | 16 | - | - | - | 28.9 | - | 49.4 | 43.0 | 36.7 | 43.0 |
| InternVL-2.5-8B [5] | 64 | - | - | - | 31.4 | 68.9 | - | - | - | 64.2 |
| Kangaroo (LLaMA2-8B) [25] | 64 | - | - | 62.7 | 30.3 | - | - | - | - | 56.0 |
| Qwen2-VL-7B [38] | 32 | 44.7 | 72.1 | 55.3 | 32.6 | 54.6 | 65.7 | 51.2 | 43.9 | 53.6 |
| Qwen2.5-VL-7B [2] | 32 | 46.2 | 73.5 | - | 33.0 | 56.3 | - | - | - | 54.5 |
| LLaVA-Video-7B [56] | 32 | 45.8 | 73.2 | 56.6 | 32.8 | 56.7 | 71.4 | 53.6 | 43.5 | 56.1 |
| Long-LLaVA-7B [41] | 32 | - | - | - | - | - | 60.3 | 51.4 | 44.1 | 52.0 |
| LLaVA-Video-72B [56] | 32 | 60.7 | 75.4 | 74.7 | 38.7 | 72.8 | 78.0 | 63.7 | 59.6 | 67.1 |
| Qwen2.5-VL-72B [2] | 34 | 62.5 | 76.8 | 75.4 | 39.3 | 73.1 | 79.4 | 71.2 | 71.8 | 74.1 |
| BIMBA-LLaVA(LLaMA3.2-8B) [15] | 64 | - | 76.9 | 60.3 | - | - | - | - | - | 50.1 |
| BIMBA-LLaVA(Qwen2-7B) [15] | 128 | 59.5 | 83.7 | 71.1 | - | 71.4 | - | - | - | 64.7 |
| VideoAgent [40] | 87 | 50.2 | 71.3 | 60.2 | - | 57.8 | 63.6 | 55.4 | 49.0 | 56.0 |
| VideoMemAgent [9] | 72 | 51.2 | 70.8 | 62.8 | - | 58.2 | 55.3 | 64.2 | 52.7 | 57.4 |
| LLaVA-Video-7B + ReAgent-V | 34 | 53.1 | 74.6 | 60.8 | 35.2 | 58.8 | 72.7 | 54.8 | 46.7 | 57.9 |
| Qwen2-VL-7B + ReAgent-V | 33 | 46.4 | 74.9 | 56.4 | 35.6 | 56.1 | 70.3 | 56.1 | 48.6 | 58.3 |
| Qwen2.5-VL-7B + ReAgent-V | 35 | 54.3 | 74.7 | 61.9 | 40.5 | 60.7 | 73.5 | 58.2 | 49.8 | 60.7 |
| LLaVA-Video-72B + ReAgent-V | 38 | 64.9 | 83.2 | 75.1 | 40.7 | 73.3 | 78.2 | 70.9 | 71.4 | 73.5 |
| Qwen2.5-VL-72B + ReAgent-V | 32 | **66.4** | **84.3** | **76.2** | **41.2** | **74.2** | **80.1** | **72.3** | **72.9** | **75.1** |

and baseline models are provided in Appendix A and B.

**Video LLM Reasoning.** This application focuses on enhancing the reasoning capability of Video LLMs using ReAgent-V. Inspired by prior work [42] that demonstrates the effectiveness of using smaller, high-quality datasets to improve LLM or VLM reasoning, we leverage ReAgent-V as a data selection mechanism to identify valuable training samples. Specifically, we apply ReAgent-V to video samples from the Video-R1-260k [10] dataset, using the scores in $E$ output by ReAgent-V during inference as the sample importance score. We retain videos and their original questions with importance scores lower than 5 (out of 10), as these cases tend to require more reflection during inference and are more challenging for the model. As such, we consider them more informative for training. The selected samples are then used to fine-tune the model using GRPO [34]. Following the experimental setup in [10, 9], we conduct comprehensive evaluations on 6 video reasoning datasets. Detailed descriptions of the benchmark datasets and baseline models are provided in Appendix A.

**VLA Alignment.** This application investigates whether ReAgent-V can go beyond improving general video understanding to effectively benefit VLA alignment. Following [57], we aim to enhance the alignment of VLA models by using ReAgent-Vas the reward model, replacing the template-based reward function originally proposed in [57]. Specifically, we apply ReAgent-V to evaluate randomly collected rollouts from OpenVLA-7B in the SIMPLER environment [23]. These rollouts are assessed across multiple dimensions—success or failure, task completeness, stability, and accuracy—resulting in an overall evaluation score. These scores are then used to align the model's behavior via trajectory-wise preference optimization (TPO) [57].

We evaluate the aligned models across four types of generalization: in-domain generalization, subject generalization, physical generalization, and semantic generalization, each consisting of multiple subtasks. Additionally, we benchmark performance against a diverse set of baselines trained under various paradigms and reward models, including OpenVLA-SFT, OpenVLA-DPO, and OpenVLA-TPO with GRAPE. All experiments are conducted under consistent settings, with each training paradigm using the same number of trajectory pairs to ensure fairness. Detailed descriptions of task setups, model configurations and hyperparameters are provided in Appendix A.

## 3.2 Main Results

We present the results of ReAgent-V across three applications: video understanding (Table 1), video LLM reasoning (Table 2), and VLA model alignment (Figure 3). Overall, ReAgent-V consistently yields substantial improvements across all tasks, achieving performance gains of 6.9%, 2.1%, and 9.8% in the respective applications. A detailed analysis for each application is provided below.

**Video Understanding.** In video understanding, ReAgent-V effectively enhances the performance of open-source LVLMs across various parameter sizes and architectures. For instance, without introduc-

Table 2: Model performance under different optimization strategies using Qwen2.5-VL-7B as the base model across various benchmarks.

| Models | Frames | Amount of data | Video Reasoning Benchmark | | | Video General Benchmark | | |
|---|---|---|---|---|---|---|---|---|
| | | | VSI-Bench | VideoMMMU | MMVU | MVBench | TempCompass | VideoMME$_{\text{w/ sub.}}$ |
| Original | 16 | - | 27.7 | 47.8 | 59.2 | 57.4 | **72.2** | 53.1 |
| SFT | 16 | 260k | 31.8 | 47.4 | 61.3 | 59.4 | 69.2 | 52.8 |
| Vanilla GRPO | 16 | 116k | 32.3 | 45.8 | 60.6 | 60.9 | 69.8 | 53.8 |
| GRPO + ReAgent-V | 16 | 52k | **33.1** | **47.9** | **63.0** | **61.4** | 70.3 | **54.2** |

ing additional frame inputs, ReAgent-V achieves average performance gains of 3.2% and 6.9% on multiple benchmarks compared to the original Qwen2.5-VL-72B and LLaVA-Video-72B, respectively. Notably, for LLaVA-Video-72B, ReAgent-V attains performance comparable to GPT-4o despite using significantly fewer input frames, highlighting its effectiveness and efficiency. Furthermore, at the same parameter scale, ReAgent-V surpasses other agentic methods such as VideoMemAgent [9] while using fewer frames and maintaining higher efficiency.

**Video LLM Reasoning.** In the video LLM reasoning, ReAgent-V outperforms vanilla GRPO by a relative average of 2.1%, while using only 45% of its training data. This demonstrates the effectiveness of data selection in enhancing video LLM reasoning, as well as the strength of ReAgent-V in guiding that selection. Moreover, ReAgent-V surpasses the SFT strategy, which was trained on the full 260k video-image dataset, achieving an average relative improvement of 4.3% across all reasoning and general benchmarks. These results collectively highlight the value of leveraging evaluation reports generated during ReAgent-V inference as rewards for guiding effective data selection.

**VLA Alignment.** In the VLA alignment task, ReAgent-V achieves a 9.8% overall improvement over the second-best baseline, GRAPE (which uses a template reward), on SIMPLER under the same setting. Specifically, on the original task, ReAgent-V outperforms GRAPE by 9.0%. These results suggest that, compared to GRAPE—which relies on a predefined reward function to evaluate trajectory quality—ReAgent-V provides more accurate rewards for the trajectories generated by VLA models, thereby achieving better alignment.

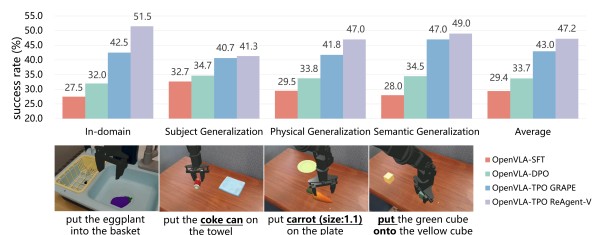

Figure 3: Comparison of ReAgent-V with OpenVLA and other reward method on the same data on the Simpler-Env environment.

## 3.3 Quantitative Analysis

In this section, we conduct comprehensive analysis to evaluate the effectiveness of incorporating entropy-calibrated frame selection and the multi-perspective reflection mechanism.

**Analysis of Frame Selection Mechanism.** In Table 3, we present the average per-sample inference time with and without frame selection, along with the corresponding performance differences across three benchmarks for various models. The results demonstrate that the frame selection strategy introduced in ReAgent-V consistently yields substantial performance gains across LVLMs with diverse architectures and parameter scales, while simultaneously reducing inference time. This highlights the effectiveness of our entropy-calibrated frame scoring strategy in selecting key frames that are both question-relevant and informative. Furthermore, we compare the performance of different video agent frameworks, and it is clear that

Table 3: Analysis on frame selection mechanism in ReAgent-V, "Time" denotes the average per-sample inference time on the corresponding benchmark.

| Model | Selection | LongBench | | VideoMME | | EgoSchema | |
|---|---|---|---|---|---|---|---|
| | | Acc.(%) | Time (s) | Acc.(%) | Time (s) | Acc.(%) | Time (s) |
| VideoAgent (GPT-4) | ✓ | 50.2 | 41.4 | 56.0 | 48.6 | 60.2 | 54.9 |
| VideoMemAgent (GPT-4) | ✓ | 51.2 | 122.3 | 57.4 | 143.7 | 62.8 | 152.6 |
| LLaVA-7B | ✗ | 52.6 | 35.99 | 56.5 | 33.07 | 57.8 | 41.96 |
| | ✓ | **53.1** | **21.38** | **57.9** | **28.38** | **60.8** | **34.86** |
| QwenVL-7B | ✗ | 46.3 | 39.53 | 60.1 | 36.52 | 61.2 | 49.23 |
| | ✓ | **46.4** | **23.02** | **60.7** | **27.33** | **61.9** | **44.08** |
| LLaVA-72B | ✗ | 64.3 | 83.24 | 74.8 | 89.37 | 73.1 | 86.53 |
| | ✓ | **64.9** | **68.19** | **75.1** | **65.38** | **73.5** | **77.53** |
| Qwen2.5-72B | ✗ | 66.3 | 88.26 | 75.9 | 81.73 | 75.0 | 87.61 |
| | ✓ | **66.4** | **69.73** | **76.2** | **68.21** | **75.1** | **79.64** |

Table 4: Reflection analysis in ReAgent-V.

| Model | Reflection | LongBench | VideoMME | EgoSchema |
|---|---|---|---|---|
| LLaVA-Video-7B | ✓ | **53.1** | **57.9** | **60.8** |
| | ✗ | 52.9 | 57.1 | 59.4 |
| Qwen2-VL-7B | ✓ | **46.4** | **58.3** | **56.4** |
| | ✗ | 46.1 | 58.2 | 55.8 |
| LLaVA-Video-72B | ✓ | **64.9** | **73.5** | **75.1** |
| | ✗ | 64.2 | 73.1 | 74.8 |
| Qwen2.5-VL-72B | ✓ | **66.4** | **75.1** | **76.2** |
| | ✗ | 65.9 | 75.0 | 75.9 |

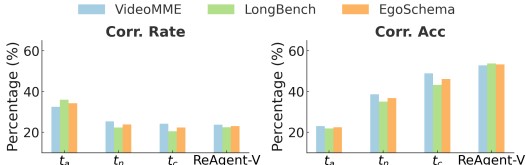

Figure 4: Comparison of four reflection strategies ($t_a$, $t_n$, $t_c$, and ReAgent-V), where Corr. Rate denotes the frequency of answer revision and Corr. Acc indicates the accuracy of those revisions.

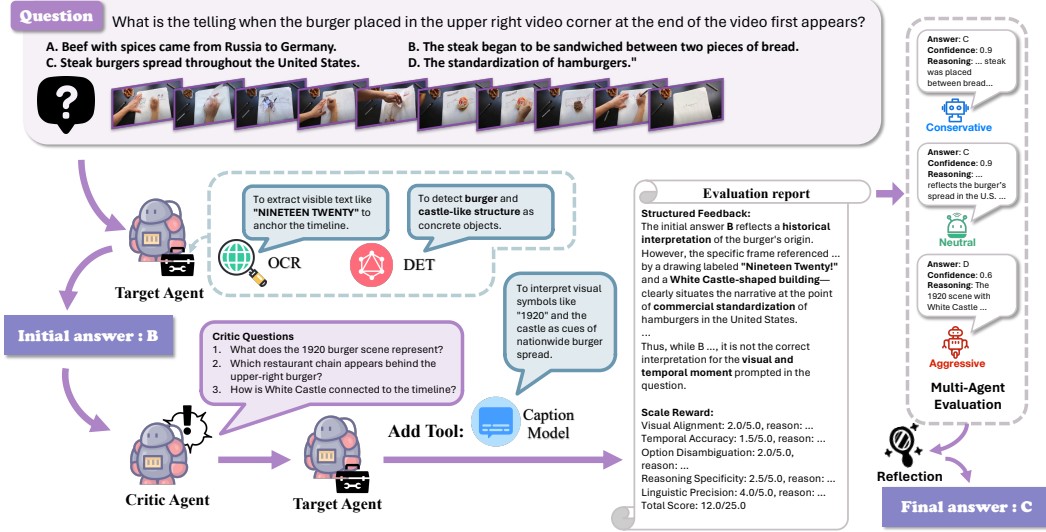

Figure 5: A case study demonstrating how ReAgent-V enhances video understanding through iterative reasoning and tool use (see additional examples in Appendix C).

ReAgent-V achieves significantly higher efficiency than other general video agent frameworks, further validating its effectiveness.

**Analysis of Reflection Mechanism.** To analyze the reflection mechanism in ReAgent-V, we conducted experiments across models with different architectures and parameter sizes. Specifically, we compare the performance of ReAgent-V with and without the reflection module on video understanding tasks. The version without reflection corresponds to the pipeline in Figure 4 where the agent directly performs tool-augmented reasoning based on selected frames. As shown in Table 4, incorporating the reflection module consistently improves the performance of various models on video understanding benchmarks. Furthermore, in Figure 4, we analyze the individual contributions of the three different perspectives - conservative, neutral, and aggressive - within the reflection process. Experimental results show that under the aggressive strategy, which allows modifying both the target agent's reasoning process and final answer, the corrected answers have the lowest accuracy. This may be because the model, in trying to follow the reflection template, mistakenly alters previously correct reasoning steps. In contrast, the conservative strategy, which only permits changing the final answer without modifying the reasoning, yields better correction performance, indicating that errors are more easily fixed when the original reasoning is preserved. By integrating three perspectives, ReAgent-V effectively mitigates errors caused by incorrect reasoning modifications while retaining the ability to correct mistakes, thereby enhancing the stability and overall accuracy of the reflection mechanism.

### 3.4 Case Study

To evaluate fine-grained video understanding, we present a case from the VideoMME dataset involving the question "What is the video conveying when the burger first appears in the upper right corner?"—a scene marked by a burger, a "1920" timestamp, and a castle-like White Castle structure as is shown in Figure 5. Initially, using OCR and object detection, the model answered B ("The steak began to

be sandwiched between two pieces of bread"), missing key symbolic and temporal cues. Reflection via three sub-questions introduced a caption model, enabling semantic interpretation that revised the answer to C, recognizing the 1920 and castle as symbols of fast food's standardization. Multi-agent evaluation showed diverse views: the aggressive agent favored D (industrialization), while neutral and conservative agents supported C. Reward scores confirmed gains in reasoning specificity and visual alignment. This case demonstrates the complexity of symbol-rich, temporally grounded video QA and underscores the value of dynamic tool integration, reflection, and reward-driven evaluation.

## 4   Related Work

**Multimodel LLMs for video understanding.** Large vision-language models (LVLMs) have been widely applied in video understanding [14, 37, 4, 21, 55, 5, 25]. These models integrate tools like Optical Character Recognition (OCR) [32], Automatic Speech Recognition (ASR) [50], video object detection [29, 17], and video captioning [36, 4, 8, 47] to better interpret complex video content. Recent work has applied these tools to tasks like video question answering by fusing visual, textual, and auditory features [30, 16]. However, earlier approaches are static—producing answers in a single pass without iterative refinement—limiting their ability to handle multi-step reasoning and self-correction [30, 16]. To overcome this, newer frameworks employ retrieval-augmented generation to dynamically select relevant video segments [35], and multi-agent systems have emerged [40, 9], enabling modality-specialized agents to collaborate for real-time refinement and error correction, mimicking human-like iterative reasoning. Reflective reasoning further helps agents adapt outputs based on new information [53].

Nevertheless, most existing methods lack real-time reward feedback and struggle with refining predictions during inference. They also tend to overprocess video frames, lacking efficient frame selection [40, 9], and often rely heavily on the model's internal capacity or require retraining for effective reflection. In contrast, our framework, ReAgent-V, introduces a unified multi-agent system with real-time, inference-aware reward signals and a novel multi-perspective reflection mechanism. This enables dynamic refinement during inference and facilitates automatic selection of high-quality outputs for fine-tuning. Consequently, ReAgent-V achieves greater adaptability, interpretability, and performance across video understanding tasks—without the need for costly annotations or static reward templates.

**Data-Centric VLM Reasoning Strategies.** To enhance reasoning accuracy, methods like supervised fine-tuning (SFT) [7] and reinforcement learning [43] with human-defined reward models such as direct preference optimization (DPO) [33] and group relative policy optimization (GRPO) [34], have been developed. These approaches rely on fine-tuning base models with large annotated datasets and reward models that prioritize human-aligned outputs [7, 33, 34]. While promising, these strategies face critical challenges: annotating high-quality temporal data, such as action sequences or causal reasoning, is labor-intensive and costly, limiting scalability. Additionally, pre-trained reward models, typically used offline, cannot adapt to a model's real-time reasoning during inference [44, 28, 34, 18]. This becomes problematic in dynamic scenarios where feedback needs to be nuanced and responsive to the evolving reasoning process [51, 19, 1]. As a result, these reward models may fail to provide timely corrections or even reinforce erroneous reasoning patterns, hindering the model's ability to improve in real-world, dynamic tasks [60]. These limitations highlight the need for more adaptive and scalable approaches to enhance model performance in complex environments.

## 5   Conclusion

We present ReAgent-V, a unified, agentic framework for video understanding that introduces inference-time reward generation and multi-perspective reflective refinement. By combining lightweight entropy-calibrated frame selection with dynamic feedback, ReAgent-V moves beyond static, single-pass reasoning and enables effective tool-augmented self-correction. Experiments on 12 datasets across three applications—video understanding, video reasoning enhancement, and vision-language-action model alignment—demonstrate consistent performance gains (up to 9.8%) with minimal overhead. We hope this framework inspires further exploration into efficient, self-improving agent systems capable of operating reliably in video understanding and reasoning.

## Acknowledgement

Y.Z., S.H. and H.Y. are partially supported by Cisco Faculty Research Award and Amazon Research Award.

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

# A  Experimental Setup

## A.1  Dataset and Baselines

### A.1.1  Video Understanding

The video understanding benchmarks evaluated in this work encompass six diverse and high-quality datasets, each meticulously curated to target a distinct and critical aspect of video-language comprehension. These benchmarks collectively span a wide range of tasks—including long-horizon reasoning, egocentric perspective analysis, multimodal alignment, and fine-grained visual-semantic understanding—thereby offering a comprehensive and rigorous evaluation of model capabilities across various real-world video scenarios.

- **LongBench [3]**: Designed to evaluate long-horizon reasoning over extended video contexts, requiring models to maintain semantic coherence across many frames.
- **NextQA [45]**: Focuses on temporal reasoning and causal understanding by presenting questions about action sequences and their logical consequences.
- **EgoSchema [31]**: A diagnostic benchmark for egocentric videos, assessing the ability to understand first-person perspectives and actions.
- **LVBench [39]**: Tailored for multi-modal alignment, especially in complex settings where audio, text, and vision interplay is critical.
- **MLVU [59]**: Emphasizes multi-task long video understanding, pushing models to perform a range of tasks such as captioning, question answering, and temporal ordering.
- **VideoMME [11]**: A comprehensive benchmark that integrates various modalities—including subtitles and OCR—and is especially suited for evaluating fine-grained visual-semantic alignment and symbolic reasoning in videos.

To rigorously assess the performance of ReAgent-V, we compare it against a broad array of baseline models that span different capabilities and design philosophies. Proprietary closed-source models include **GPT-4o [14]**, known for its advanced multimodal reasoning, and **Gemini-1.5-Pro [37]**, optimized for long-context multimodal input. Among open-source video-language models, the evaluation includes:

- **ShareGPT4Video [4]**: Enhances caption-driven video question answering by integrating ShareGPT-style conversational supervision into multimodal video training. It leverages large-scale user-generated captions and dialogues to align vision and language effectively, enabling better generalization to open-ended video queries.
- **VideoChat2 [21]**: A chat-centric video-language model built to support multi-turn, conversational interactions grounded in video content. It emphasizes interactive understanding, with capabilities for dialogue continuity, temporal grounding, and multi-modal alignment, making it ideal for assistive agents and education scenarios.
- **LLaVA-Video [54]**: An extension of the LLaVA framework adapted for video inputs, available in both 7B and 72B parameter versions. It employs frame-wise and temporal fusion techniques to convert video sequences into language-aligned embeddings, demonstrating strong performance on visual question answering and summarization tasks.
- **Qwen2 [2]** and **Qwen2.5-VL [2]** families: Developed by Alibaba, these models exhibit high performance on both image and video understanding benchmarks through advanced multi-modal alignment. Qwen2.5-VL particularly excels in handling long-context and dense visual-textual reasoning tasks, supported by a unified visual-language architecture.
- **InternVL-2.5 [6]**: A high-performing open-source model that scales both the architecture and training data to improve generalization. It incorporates vision-language alignment techniques, dense region grounding, and optimized pretraining routines to support both short and long video tasks with high efficiency.
- **BIMBA-LLaVA [15]**: Introduces a selective scan compression mechanism tailored for long videos, where it prunes redundant frames while preserving key semantic content. This compression-aware training improves inference speed and memory usage while maintaining accuracy on temporal and contextual reasoning tasks.

- **Kangaroo [25]**: Designed specifically for long-context video input, Kangaroo adopts hierarchical attention and memory-efficient tokenization strategies to process hundreds of frames. It is particularly effective for document-level video understanding, such as meeting summarization or sports analysis.

- **Long-LLaVA [41]**: A variant of LLaVA optimized for efficient multi-frame processing. It integrates temporal coherence constraints and cross-frame attention mechanisms, making it capable of capturing nuanced motion patterns and temporal dependencies for improved video QA and description.

- **VideoAgent [40]**: An agent framework for long-form video understanding that mimics human cognition through LLM-guided reasoning, CLIP-based retrieval, and VLM-driven state updates.

- **VideoMemAgent [9]**: An agent framework for structured long-form video understanding that integrates unified temporal and object memory with tool-augmented reasoning, enabling multi-round chain-of-thought inference across complex video content.

Across all these baselines, **ReAgent-V** almost outperforms in terms of accuracy, interpretability, and computational efficiency, largely due to its entropy-guided frame selection and multi-perspective reflection mechanism that support real-time reward-driven answer refinement.

Experiments are run on two H100-96GB GPUs with NVLink, using 64 frames per video. CLIP (ViT-L/14-336) is used for vision-language alignment. Videos are processed using decord and ffmpeg, with audio chunked for transcription. At inference, ReAgent-V performs tool-augmented reasoning by dynamically invoking OCR, ASR, DET and other modules. A multi-agent reflection strategy—comprising conservative, neutral, and aggressive evaluators—is followed by a meta-agent decision step.

### A.1.2 Video LLM Reasoning

To investigate how ReAgent-V can enhance the reasoning capability of Video LLMs, we follow the experimental setup from [10] and apply ReAgent-V to filter the video portion of the Video-R1-260k dataset. Specifically, during inference, we retain samples with importance scores lower than 5 (out of 10) as these tend to be more challenging and thus more valuable for training. We then fine-tune the Qwen2.5-VL model on 8 NVIDIA H100 80GB GPUs using the filtered dataset and compare it against several baseline models defined in the paper. Our results show that the model achieves a 2.1% improvement in overall performance while using only 45% of the original training data, demonstrating the effectiveness of high-quality sample selection in enhancing video reasoning.

Benchmarks. We evaluate our model on six video benchmarks: VSI-Bench [12], VideoMMMU [13], MMVU [58], MVBench [22], TempCompass [27], and VideoMME [11]. The first three benchmarks focus primarily on video reasoning tasks that assess the model's ability to understand and reason over complex video semantics, while the latter three are general video understanding benchmarks involving a mix of perception and reasoning challenges. For MMVU, we evaluate using its multiple-choice question subset to ensure stability and consistency.

### A.1.3 VLA Alignment

To construct a preference-based dataset for aligning vision-language-action (VLA) models, we collect and process rollouts from the SIMPLER environment [23]. Specifically, we construct a reward-ranked dataset for VLA alignment by sampling trajectories from four original in-domain tasks using the OpenVLA-7B-SFT-Simpler (OpenVLA-SFT) model, a supervised-finetuned baseline released by [57], with five rollouts per task. The remaining three generalization tasks are kept unseen and excluded from alignment. After data collection, we use ReAgent-V to evaluate each trajectory segment, assigning scalar reward scores based on task success, stability, completeness, and accuracy. For each task, we sample 20 best-vs-worst trajectory pairs, matched by task type and initial environment state. This yields 80 total trajectories, which are then partitioned into two RLDS-formatted datasets: a chosen set containing high-reward trajectories and a rejected set with low-reward counterparts.

We conduct all VLA alignment training on a single NVIDIA A100 GPU (80GB). The model is initialized from OpenVLA-SFT. Training employs LoRA (rank=32) with a learning rate of 2e-5 and dropout of 0.0. We use gradient accumulation with a step size of 4 to simulate larger batch sizes under

limited GPU memory. After training, the base model is merged with LoRA weights to obtain the final model. Our alignment process follows the Trajectory-wise Preference Optimization (TPO) [57] paradigm, where the objective is to learn a reward-aligned policy using the preference pairs described above. All results reported in the main paper are obtained after 6,800 steps of alignment training.

We evaluate all VLA models in the SIMPLER environment, covering four types of generalization: in-domain, subject generalization, physical generalization, and semantic generalization. In-domain consists of the four original tasks in the SIMPLER environment. Subject generalization includes three new tasks involving objects not seen during training. Physical generalization comprises eight tasks created by varying the width, height, and size of known objects. Semantic generalization includes four tasks where the original prompts are rephrased with synonymous instructions. Each taskset comprises several subtasks, and each subtask is evaluated over a fixed number of distinct episodes. The final metric is the average task success rate across episodes. For all baseline models—OpenVLA-SFT, OpenVLA-DPO, and OpenVLA-TPO with GRAPE—we use results reported by [57]. Our evaluation strictly follows the same experimental setup, adopting identical task definitions and environmental configurations to ensure a fair and consistent comparison.

# B   Evaluation Criteria and Prompts

## B.1   Tool Selection Process

---

**Tool Selection Prompt**

**[Task]**
Carefully analyze the video content and identify exactly what information needs to be retrieved to support answering the given question. To answer the question step by step, list all the physical entities related to the question you want to retrieve, you can provide your retrieve request to assist you by the following JSON format:

**[Tool Functions]**
The Tool Factory supports the following types of retrieval via specialized tools:

- **Text Extraction:**   Tools: `OCR`, `ASR` — extract embedded text (e.g., signs, timestamps) and transcribe speech or narration.
- **Object Detection and Grounding:**   Tool: `Grounding DINO` — detects objects and aligns them with natural language prompts.
- **Multimodal Matching:**   Tools: `CLIP` — perform image-text alignment and retrieval based on semantic similarity.
- **Structured Scene Understanding:**   Tools: `Scene Graph`, `Action Detector` — construct structured graphs of object relations and detect visual actions.
- **Video Reasoning:**   Tools: `ShareGPT4Video`, `VQA Model` — enable video-based question answering and explanation generation.
- **Caption Generation:**   Tool: `Captioning Model` — generate scene-level textual descriptions.
- **Emotion and Identity Recognition:**   Tools: `Face Recognition`, `Emotion Detector` — detect character identities and their emotional states.

**[Output format]:**
Use the following format to list the functional categories and corresponding tools required:

```
[
  {"function": "Function Category",
  "tools": ["Tool1", "Tool2"]}
]
```

**Example 1:**
Original question: "How many blue balloons are over the long table in the middle of the room at the end of this video?"
Options: "A. 1. B. 2. C. 3. D. 4."
Output:

---

```
[
  {"function": "Visual Object Detection",
   "tools": ["Grounding DINO"]},
  {"function": "Structured Scene Understanding",
   "tools": ["Scene Graph"]},
  {"function": "Numerical Reasoning",
   "tools": ["ShareGPT4Video"]}
]
```

**Example 2:**
Original question: "In the lower left corner of the video, what color is the woman wearing on the right side of the man in black clothes?"
Options: "A. Blue. B. White. C. Red. D. Yellow."
Output:

```
[
  {"function": "Visual Object Detection",
   "tools": ["Grounding DINO"]},
  {"function": "Structured Scene Understanding",
   "tools": ["Scene Graph"]},
  {"function": "Global Scene Description",
   "tools": ["Captioning Model"]}
]
```

**Example 3:**
Original question: "In which country is the comedy featured in the video recognized world-wide?"
Options: "A. China. B. UK. C. Germany. D. United States."
Output:

```
[
  {"function": "Audio Text Extraction",
   "tools": ["ASR"]},
  {"function": "Global Semantic Summary",
   "tools": ["Captioning Model"]},
  {"function": "Commonsense Reasoning",
   "tools": ["ShareGPT4Video"]}
]
```

**Example 4:**
Original question: "Describe what the chef does to prepare the pasta dish in the video."
Options: ""
Output:

```
[
  {"function": "Audio Instruction Extraction",
   "tools": ["ASR"]},
  {"function": "Object and Action Detection",
   "tools": ["Grounding DINO", "Action Detector"]},
  {"function": "Structured Scene Understanding",
   "tools": ["Scene Graph"]},
  {"function": "Narrative Generation",
   "tools": ["Captioning Model"]}
]
```

**[Now begin]**
Note that you don't need to answer the question in this step, so you don't need any infomation about the video of image. You only need to provide your retrieve request (it's optional), and I will help you retrieve the infomation you want. Please provide the json format.

This section introduces the Tool Selection Prompt framework, designed to guide retrieval-aware video question answering through structured tool invocation. By decomposing a user question into

functional information needs—such as object detection, text extraction, or reasoning—the prompt enables precise specification of which tools (e.g., OCR, Grounding DINO [26], CLIP, Scene Graph, ASR) should be employed. The unified JSON output format supports downstream automation and tool chaining, ensuring modularity and extensibility. Multiple illustrative examples demonstrate how diverse question types can be mapped to appropriate retrieval functions and tools, from counting objects to recognizing emotional states or extracting narrated instructions. Notably, if additional visual tools or other modalities are required, users can simply modify the tool selection section to include the necessary tools, making this prompt a highly flexible and scalable interface for building robust multimodal QA pipelines.

## B.2 Reflection Prompt

The Reflection Prompt framework is divided into three complementary strategies - Neutral B.2.1, Conservative B.2.2, and Aggressive B.2.3 - each offering a different level of intervention to revise initial video QA outputs. These prompts are designed to support multi-stage reasoning and grounded visual correction by reassessing object perception, answer validity, or full reasoning chains. Depending on the task format (e.g., open-ended QA, multiple choice), scoring mechanism (e.g., scalar reward, structured feedback), or desired reflection granularity, users can select the appropriate strategy: Neutral for perceptual correction, Conservative for stability-focused validation, and Aggressive for complete answer reconstruction. These templates can also be adapted or hybridized to support diverse evaluation workflows, enabling flexible deployment across multi-task QA pipelines.

### B.2.1 Neutral Reflection Prompt

---

**Entity-Centric Revision**

**[Task]**
You are a neutral agent responsible for reassessing the initial model answer by correcting only visual misperceptions of scene elements. Your role is to revise the perceptual input (i.e., object/entity grounding) while preserving the original reasoning logic. Do not introduce any new reasoning steps.

**[Reflection Requirement]**
If the original answer is based on a misidentified visual entity, correct that grounding (e.g., object type, color, spatial position). Keep the interpretation process unchanged. This strategy focuses on refining *what is seen*, not *how it is reasoned*.

**[Reflection Procedure]**

1. Re-examine the video or image frames for **object-level details** (e.g., people, objects, colors, gestures).

2. Determine whether the initial answer failed due to incorrect or missing perception of visual entities.

3. Adjust the relevant scene elements accordingly (e.g., update object color, position, or identity).

4. Reuse the original reasoning chain, now applied to the corrected visual grounding.

**[Evaluation Guidelines]**

- Remain neutral — do not assume the initial answer is incorrect unless there is clear evidence of perceptual error.

- Do not modify the reasoning logic — preserve the original inference path.

- Only revise the interpretation of visual content (i.e., what objects appear and their attributes).

**[Input]**

- Question: {text}

- Initial Answer: {initial_answer}

- Eval Report: {structured feedback or scalar reward}

---

> **[Output]**
> Use the following format to provide the final revised answer after entity-level correction.
> Only output the revised answer and its confidence score — no explanations, no justifications,
> and no extra text of any kind.
> ```
> Final Answer:  ({free-form revised answer}, confidence score:  0-1)
> ```

### B.2.2 Conservative Reflection Prompt

> **Answer-Focused Revision**
>
> **[Task]**
> You are a conservative agent responsible for validating the initial model answer. Your role
> is to preserve the original answer unless there is irrefutable visual evidence that directly
> contradicts it. Do not revise or reinterpret the reasoning or scene elements unless the
> contradiction is absolute.
>
> **[Reflection Requirement]**
> Only revise the final answer itself—do not modify scene elements or reasoning logic. If
> the original answer remains potentially valid, it should be retained. This strategy aims for
> minimal intervention: conservative reflection focuses on maintaining output stability unless
> overwhelming visual contradiction is present.
>
> **[Reflection Procedure]**
>
> 1. Re-examine the visual content for any direct contradiction to the initial answer.
> 2. Accept only literal, unambiguous visual cues that fully invalidate the original answer.
> 3. Do not alter object interpretation, scene structure, or logical flow.
> 4. Revise the final output only if the contradiction is undeniable and renders the
>    original answer unsupportable.
>
> **[Evaluation Guidelines]**
>
> - Retain the original answer if:
>     - Any uncertainty or ambiguity exists in the evidence
>     - Visual information lacks a clear, literal contradiction
>     - A reasonable observer could still accept the original answer
> - Revise the answer only if:
>     - The visual content presents overwhelming and explicit contradiction
>     - The revised answer exactly matches visible evidence without requiring inter-
>       pretation
>     - The contradiction is strong enough to convince any neutral evaluator
>
> **[Input]**
>
> - Question: {text}
> - Initial Answer: {initial_answer}
> - Eval Report: {structured feedback or scalar reward}
>
> **[Output Format]**
> Use the following format to provide the final decision after conservative validation. Only
> output the revised answer and your confidence score—no explanations, no justifications, and
> no extra text of any kind.
> ```
> Final Answer:({free-form revised answer}, confidence score:  0-1)
> ```

### B.2.3 Aggressive Reflection Prompt

**Reasoning-Driven Revision**

**[Task]**
You are an aggressive agent responsible for actively challenging the initial model answer. Your role is to revise both the reasoning process and the visual understanding in order to reconstruct a superior alternative.

**[Reflection Requirement]**
This strategy requires modifying both the reasoning steps and the associated scene entities. It involves the widest scope of change and is intended to completely overturn the original logic and rebuild a more accurate answer from scratch. Accept loose semantic alignment, reinterpret ambiguous scenes, and prioritize alternative perspectives over the original.

**[Reflection Procedure]**

1. Re-examine all video frames for cues that could support a different interpretation.
2. Modify the understanding of relevant visual entities (objects, attributes, spatial relations).
3. Reconstruct the reasoning chain based on the newly grounded observations.
4. Select a new answer that best fits the revised reasoning and evidence.

**[Evaluation Guidelines]**

- Always replace the original answer — the default assumption is that it is suboptimal or incorrect.
- Consider alternative answers even when based on partial, ambiguous, or abstract cues.
- Allow semantic flexibility (e.g., "canine" = "dog", "liquid" = "water") and recontextualization.
- Prioritize comprehensive reinterpretation to generate an improved final response.

**[Input]**

- Question: `{text}`
- Initial Answer: `{initial_answer}`
- Eval Report: `{structured feedback or scalar reward}`

**[Output Format]**
Use the following format to provide the final revised answer after full reasoning and entity-level revision. Only output the revised answer and your confidence score — no explanations, no justifications, and no extra text of any kind.
`Final Answer: ({free-form revised answer}, confidence score: 0-1)`

### B.2.4 Overall Reflection Prompt

**Multi-Perspective Reflection Aggregation**

**[Task]**
You are a specialized Meta-Agent for video question answering. Your role is to integrate the answers and confidence scores from three agents with different reflection strategies. Your goal is to synthesize a final answer by evaluating answer quality, confidence levels, and semantic overlap.

**[Multi-Perspective Inputs]**

- Conservative Agent (Answer-Focused Reflection)
  Answer: `{answer_conservative}`, Confidence: `{conf_conservative}`
- Neutral Agent (Entity-Centric Reflection)
  Answer: `{answer_neutral}`, Confidence: `{conf_neutral}`

- Aggressive Agent (Reasoning-Driven Reflection)
  Answer: {answer_aggressive}, Confidence: {conf_aggressive}

**[Decision Procedure]**

Step 1 — High-Confidence Fusion

If all three confidence scores exceed their respective thresholds:

- `conf_conservative` $\geq 0.6$
- `conf_neutral` $\geq 0.7$
- `conf_aggressive` $\geq 0.8$

Then:

- Combine the three answers {answer_aggressive}, {answer_aggressive}, {answer_aggressive}
- Extract shared components and consistent semantic information
- Remove contradictions or unsupported segments
- Produce a final, coherent free-form answer that integrates the common insights

Step 2 — Confidence-Based Selection If one or more confidence scores fail to meet their thresholds:

- Select the answer with the highest confidence score among the three agents
- Use that agent's full revised answer as the final output

**[Evaluation Criteria]**

- **Semantic Overlap:** Identify key phrases, facts, and themes that appear in multiple answers
- **Contradiction Removal:** Discard any segments that directly conflict with others
- **Fluency:** Ensure the final answer reads as a natural, well-formed sentence

**[Input]**

- Question: {text}
- Initial Answer: {initial_answer}
- Agent Answers:
  - {answer_conservative}
  - {answer_neutral}
  - {answer_aggressive}
- Agent Confidences:
  - {conf_conservative}
  - {conf_neutral}
  - {conf_aggressive}

**[Output Format]**

Use the following format to provide the final revised answer. Only output the revised answer — no explanations, no justifications, and no extra text of any kind.

`Final Answer: ({free-form revised answer}`

## B.3 Critical Prompt

This section introduces two complementary evaluation prompts—Clarification Question Generation B.4 and Eval Report Generation B.5 - designed to support critical diagnosis of model answers in video question answering. The Clarification Prompt generates precise, targeted sub-questions when an answer is incomplete or ambiguous, uncovering reasoning or grounding gaps by referencing specific visual or contextual elements. It is particularly useful for both open-ended and multiple-choice QA tasks, helping localize errors without requiring immediate scoring. The Eval Report Prompt provides a structured scoring mechanism across five dimensions—visual alignment,

temporal accuracy, option disambiguation, reasoning specificity, and linguistic precision—yielding both qualitative feedback and a scalar reward useful for leaderboard tracking, model tuning, or reinforcement learning. Together, these prompts enable a flexible and task-adaptable evaluation framework: clarification questions are ideal for diagnosing failures and guiding revisions, while scalar scores support performance benchmarking. Depending on the evaluation goal—error localization, answer justification, or progress tracking—these prompts can be selectively applied or adapted, with clarification prioritized in open-domain QA and scoring emphasized in competitive or quantitative settings.

## B.4 Critical Question Prompt

---

**Clarification Questions Generation Prompt**

**[Task]**
You are a critic agent tasked with evaluating the quality of the initial answer $A_0$ generated by the target agent, based on the provided question, context, and video content. If the answer is deemed unsatisfactory, your goal is to help localize potential errors by generating one or more sub-questions. These sub-questions mustshould be highly specific and firmly grounded in the visual or contextual evidence.

**[Input Data]**
Question: "{text}"
Answer: "{answer}"
Context: "{context}"

**[Evaluation Criteria]**

1. Check if the answer fully addresses the question
2. Verify all key elements from context are included
3. Assess whether video content supports the answer
4. If not, raise sub-questions to expose missing or uncertain reasoning

**[Clarification Guidelines]**
If the answer is incomplete, generate 1–3 ultra-specific clarification questions following these rules:

- Must start with: "What", "Where", "When", "Which", or "How"
- Must reference concrete elements from context/video
- No vague pronouns ("it", "they") — use specific nouns
- Examples: "Which timestamp shows the error?" or "How many frames were processed?"

**[Output Format]**

```
- Return []                              (for complete answers)
- Return ["question1?", "question2?", ...] (for incomplete answers)
```

---

### B.5 Evaluation Report Prompt

---

**Eval Report Generation**

**[Task]**
You are a critic agent tasked with evaluating the quality of the initial answer $A_0$ generated by the target agent, using the given question and contextual information. Your goal is to provide structured diagnostic feedback by scoring the answer across multiple dimensions and computing a final scalar reward (0.0–1.0) based on the total score.

**[Input Data]**

- Question: `{text}`
- Context: `{context}`
- Initial Answer: `{initial_answer}`

**[Evaluation Criteria]**
Rate the answer on the following five dimensions (0.0–5.0 scale for each):

- **Visual Alignment**: Is the answer aligned with visible video evidence?
- **Temporal Accuracy**: Is the answer consistent with the timeline or timestamps?
- **Option Disambiguation**: If multiple options are similar, does the answer clearly justify the selected one?
- **Reasoning Specificity**: Is the reasoning clear, focused, and appropriately detailed?
- **Linguistic Precision**: Is the answer grammatically correct and semantically accurate?

**[Output Format]**
Return a JSON object with detailed scoring and reasons for each dimension, plus the final total and normalized scalar reward:

```
{
  "scores": {
    "visual_alignment": {"value": float (0.0-5.0), "reason": "..."},
    "temporal_accuracy": {"value": float, "reason": "..."},
    "option_disambiguation": {"value": float, "reason": "..."},
    "reasoning_specificity": {"value": float, "reason": "..."},
    "linguistic_precision": {"value": float, "reason": "..."}
  },
  "total_score": float (0.0-25.0),
}
```

---

### B.6 Implementation Workflow of ReAgent-V

Figure 6 illustrates the complete inference workflow of ReAgent-V, a modular agent system designed for video-based question answering. The pipeline begins with unified initialization via `load_default`, followed by ECRS-based keyframe selection to reduce redundancy while preserving semantic relevance. A dictionary-based tool selection mechanism dynamically activates symbolic extraction tools (e.g., OCR, ASR, DET) based on the input query. Extracted textual context is merged into `modal_strings` and composed into a multimodal prompt for LLaVA-based initial answering. If critical gaps are identified, the system enters a reflective reasoning stage to revise the answer. Finally, an evaluation report is generated to assess the quality of both the initial and refined responses.

## C More Visualization Results

```
ReAgent-V Inference Pipeline

>>> from init_modules import *
>>> qa_system = ReAgentV.load_default("path/to/base_model")
>>> frames = qa_system.load_video_frames("example.mp4", max_frames)
>>> key_frames, key_indices = qa_system.ECRS_select_keyframes(frames,
    question)
>>> # Pass custom tool list to dynamically control which tools to apply and
    revise the tool selection prompt template accordingly.
>>> tool_list = ["OCR", "ASR", "DET", "SceneGraph", "Grounding DINO", "
    Caption Model", "CLIP", "ShareGPT4Video", "VQA Model", "Action Detector",
     "Face Recognition", "Emotion Detector"]
>>> # "selected_tools" is a boolean map indicating required tools for
    answering the question, e.g., {"OCR": True, "ASR": False, "DET": False,
    ...}
>>> selected_tools = qa_system.select_tools(question, tool_list=tool_list)
>>> # Use selected_tools to use only necessary models and extract tool-
    specific information from key_frames into modal_info, a dictionary keyed
    by tool name.
>>> modal_info = qa_system.extract_modal_info(key_frames, question, **
    selected_tools)
>>> # Build a multimodal prompt by integrating tool-specific information from
     modal_info into the question template.
>>> prompt = qa_system.build_multimodal_prompt(question, modal_info,
    key_indices, len(key_frames))
>>> initial_answer, = qa_system.model_inference(prompt, key_frames)
>>> # Critic-driven refinement if necessary
>>> critical_qs = qa_system.generate_critical_questions(question,
    initial_answer, modal_info, key_frames)
>>> if critical_qs:
...     updated_infos = {}  # {q_i: {tool_name: info}}, mapping each critic
    question to its tool-specific info.
...     for q_i in critical_qs:
...         tools_i = qa_system.select_tools(q_i, tool_list=tool_list)
...         info_i = qa_system.extract_modal_info(key_frames, q_i, **tools_i)
...         updated_infos[q_i] = info_i

...     # Wrap the original modal_info into {question: modal_info}
...     context_infos = {question: modal_info, **updated_infos}  # Merge all
    into a unified context dict

...     report = qa_system.generate_eval_report(question, initial_answer,
    context_infos, key_frames)
...     # get_reflective_final_answer applies three reflection strategies (
    neutral, aggressive, conservative) and merges the best answer using the
    overall_prompt_template.
...     final_answer = qa_system.get_reflective_final_answer(
...         question, initial_answer, report, key_frames)
>>> else:
...     final_answer = initial_answer
```

Figure 6: `ReAgent-V` inference pipeline: after ECRS keyframe selection, tools are dynamically selected to generate an initial answer. If critical questions arise, tool outputs are updated for reflection. Three reasoning strategies are used to revise or confirm the answer.

## C.1 Visualization of Frame Selection

These case studies qualitatively demonstrate the effectiveness of ECRS frame selection compared to uniform sampling across a range of video question answering scenarios. In each example, ECRS consistently captures frames that are more semantically aligned with the question, providing richer context and clearer evidence to support reasoning. For instance, in Figure 7, ECRS selects

**Four Frames Sampled Using Uniform Sampling**

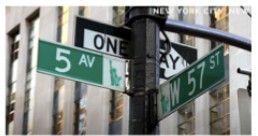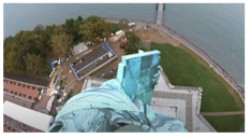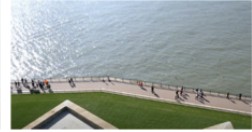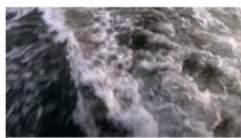

**Four Frames Sampled Using ECRS Sampling**

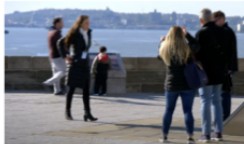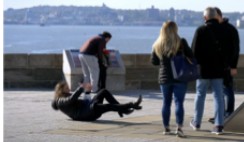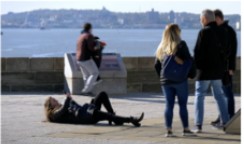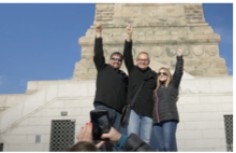

**Question about the video : How did the photographer who took pictures of the three tourists in the video take the photo?**

Figure 7: ECRS sampling better captures the interaction between the photographer and the tourists.

**Four Frames Sampled Using Uniform Sampling**

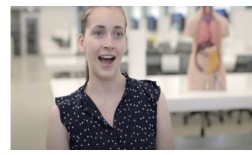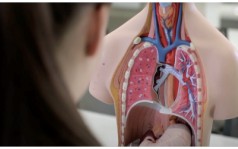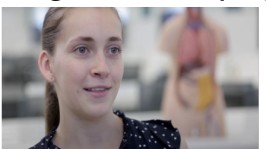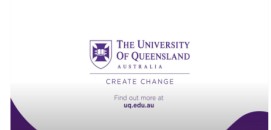

**Four Frames Sampled Using ECRS Sampling**

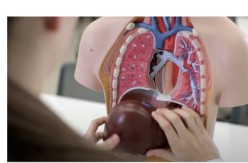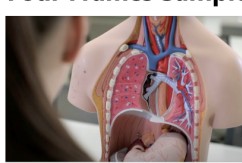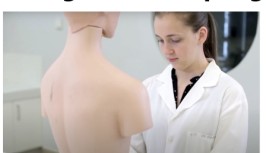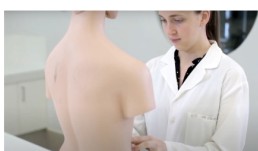

**Question about the video : What organ did the woman in the video remove from the medical model ?**

Figure 8: ECRS highlights the woman's action of removing the organ more clearly than uniform sampling.

frames that highlight the interaction between the photographer and tourists, directly addressing the question about how the photo was taken—unlike uniform sampling, which includes generic landscape shots. In Figure 8, ECRS captures the precise moment when the woman interacts with the medical model, effectively grounding the answer to the question about organ removal. Similarly, Figures 9 and 10 show that ECRS preserves critical moments involving text and actions—such as the student explaining his motivation for eating the banana or the banana-taping act that contextualizes the replaced painting—while uniform sampling misses or misaligns with these moments. Across all examples, ECRS provides temporally and semantically focused evidence that improves alignment between selected frames and the target question, validating its superiority in supporting visual reasoning. Note that although ECRS typically selects 20–40 informative frames per video for downstream processing, only four representative frames are visualized here by evenly sampling from the selected set, in order to maintain clarity and consistency in comparison.

## C.2 Visualization of Evaluation Report

These four evaluation report cases demonstrate how ReAgent-V leverages visual tools and multi-agent collaboration to refine or correct initial answers by grounding them more accurately in visual and textual evidence. In Figure 11, the system correctly identifies the Dragon Boat Festival as the main theme of the video by combining OCR and DET outputs to interpret cultural symbols and scene elements. Figure 12 shows a quantitative reasoning adjustment, where the agent detects two birds and

**Four Frames Sampled Using Uniform Sampling**

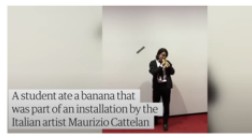 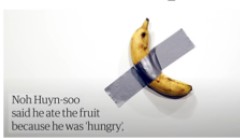 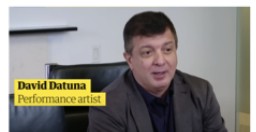 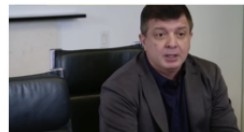

**Four Frames Sampled Using ECRS Sampling**

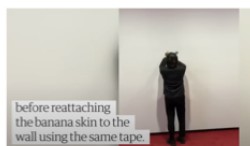 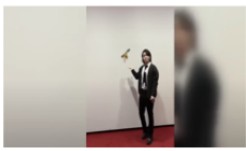 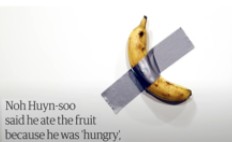 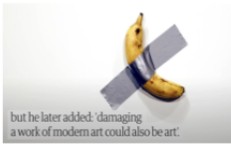

**Question about the video : Based on the video, which of the following descriptions the reason why the student ate the banana?**

Figure 9: ECRS sampling emphasizes the reason behind the student eating the banana with supporting text.

**Four Frames Sampled Using Uniform Sampling**

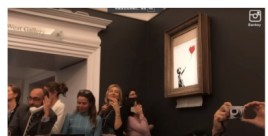 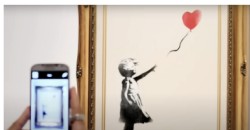 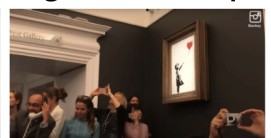 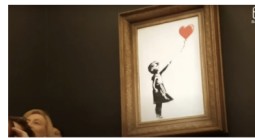

**Four Frames Sampled Using ECRS Sampling**

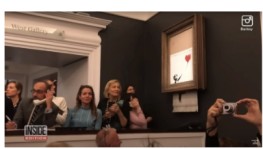 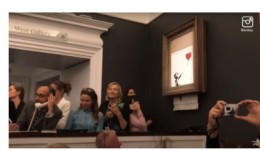 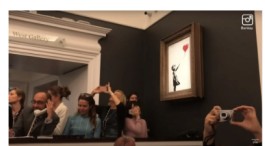 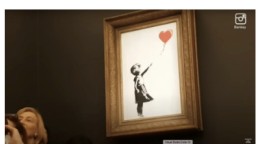

**Question about the video : Which elements are depicted in the painting introduced by the video?**

Figure 10: ECRS captures the banana-taping act that replaced the shredded painting, providing better context.

revises the initial bird count from three to two, aligning the answer with grounded visual analysis. In Figure 13, the model originally misidentifies the cat's color, but after grounding the detected regions and reviewing visual attributes, the agents collaboratively revise the answer to the correct "orange and white." Finally, Figure 14 illustrates a geographic correction where the system initially mislabels the terrain as "tropical," but after integrating CLIP-based semantics and detailed grounding, it updates the answer to "polar," aligning better with snowy and icy visual cues. These examples highlight how reflective critique and grounded evidence improve factual accuracy and visual-textual alignment.

## C.3  Visualization of VLA Alignment

Figure 15 visualizes six robotic manipulation tasks and highlights the progression from failure to success after applying reflection-guided reward correction during policy fine-tuning. In each task—carrot placement, spoon placement, eggplant basket placement, cube stacking, coke can placement, and sprite can placement—the left sequence illustrates the robot's initial failure to complete the action correctly, while the right sequence demonstrates the corrected behavior after fine-tuning. The visualizations show that the robot learns to adapt its positioning, trajectory, and precision based on reflective feedback. For example, in the carrot and spoon placement tasks, the

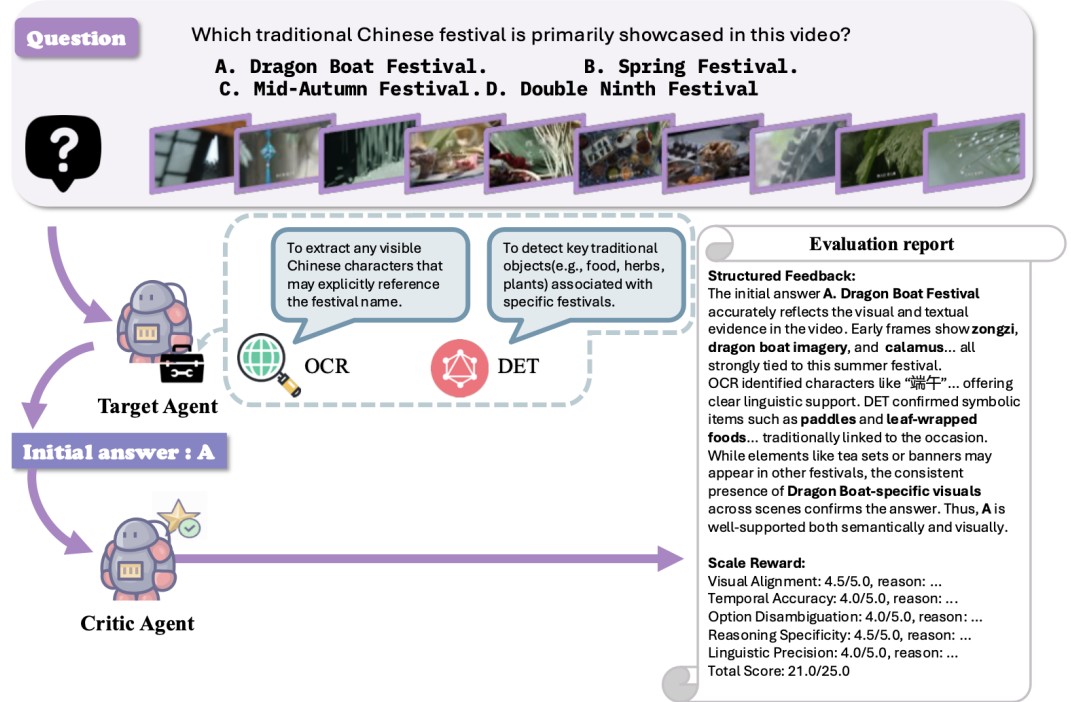

Figure 11: Visual and textual cues confirm that the video primarily showcases the Dragon Boat Festival.

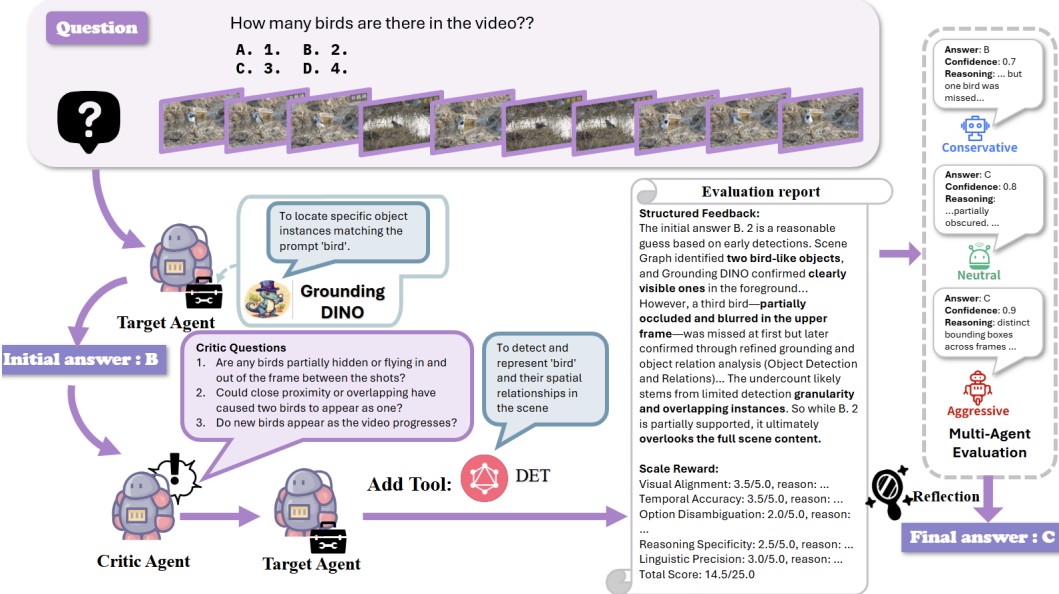

Figure 12: Multimodal analysis revises the bird count in the video from two to three after detecting an occluded bird.

gripper initially misaligns with the target location but is corrected to center the object on the plate or towel. In the eggplant and cube stacking tasks, the robot improves its grasp and drop accuracy, successfully placing the object in or on the intended target. In the coke and sprite can tasks, the robot adjusts its vertical alignment and release timing to ensure stable placement. These results collectively demonstrate that reflection-driven fine-tuning enhances the robot's task completion reliability across diverse object manipulation scenarios.

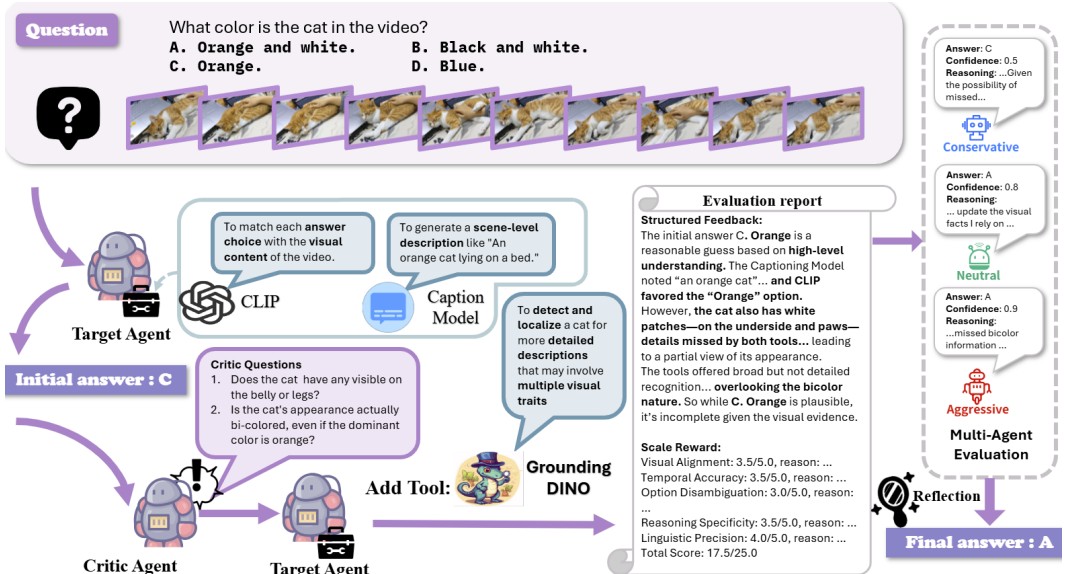

Figure 13: Multi-agent collaboration corrects the cat's color in the video from "Orange" to "Orange and white."

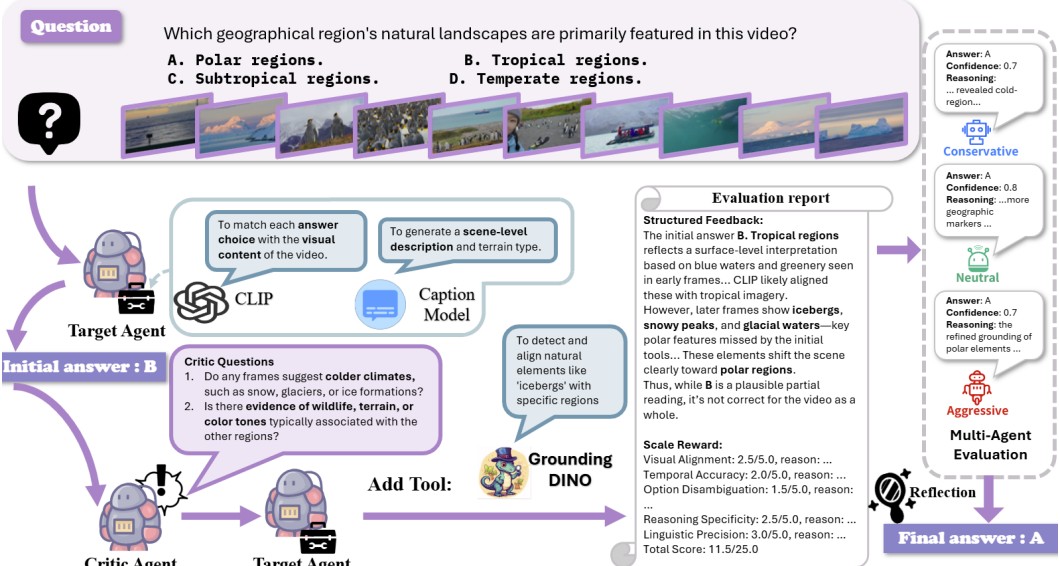

Figure 14: Agents identify icebergs and snowy terrain, correcting the geographical region from "Tropical" to "Polar."

# D    Additional Experimental Results

## D.1    Trends of Selection Scores Across Videos

Figure 16 illustrates the selection behavior of ECRS across various videos by comparing its score distribution with CLIP and Entropy-based baselines. The red bars indicate that ECRS consistently selects frames near local peaks, capturing semantically or visually informative moments. Compared to CLIP and Entropy score selections, ECRS demonstrates more temporally clustered choices that

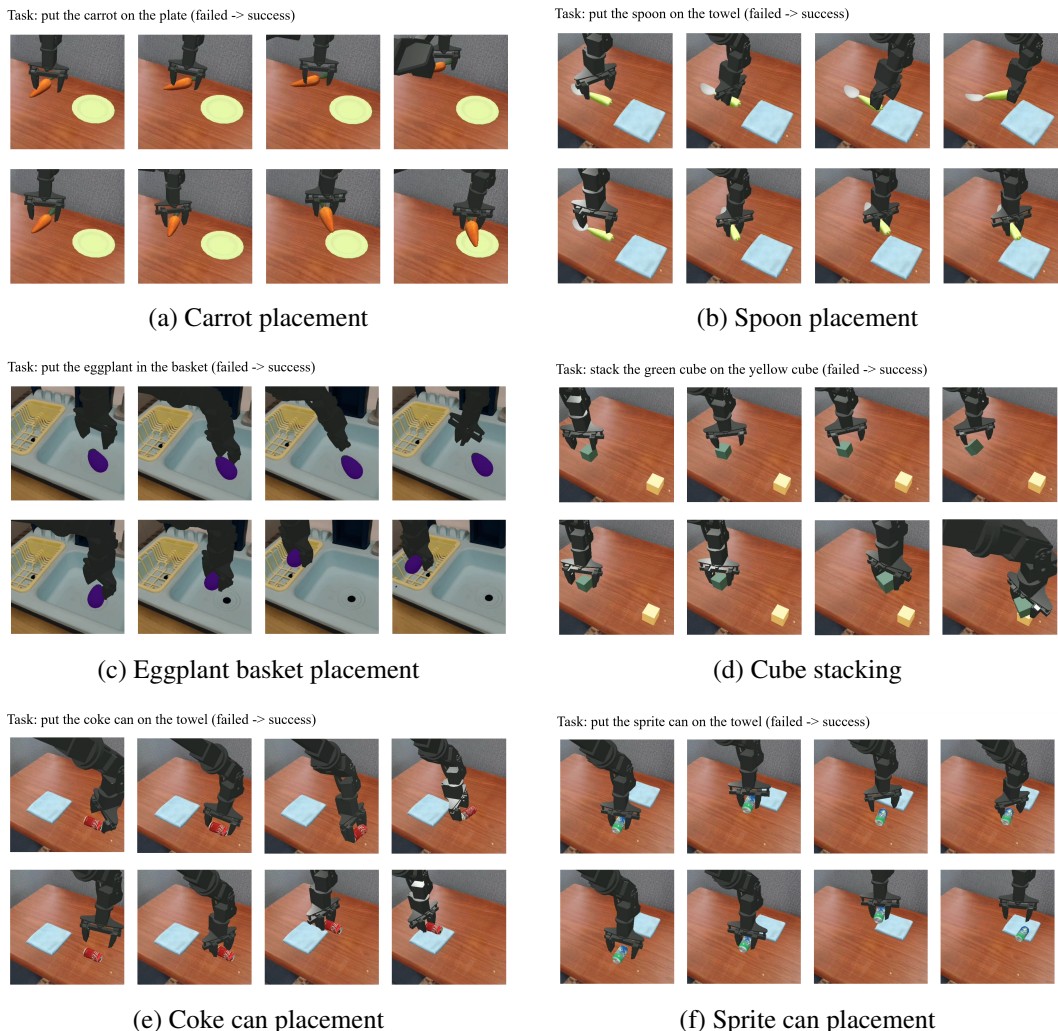

Task: put the carrot on the plate (failed -> success)

Task: put the spoon on the towel (failed -> success)

(a) Carrot placement

(b) Spoon placement

Task: put the eggplant in the basket (failed -> success)

Task: stack the green cube on the yellow cube (failed -> success)

(c) Eggplant basket placement

(d) Cube stacking

Task: put the coke can on the towel (failed -> success)

Task: put the sprite can on the towel (failed -> success)

(e) Coke can placement

(f) Sprite can placement

Figure 15: Visualization of robotic task executions before and after policy fine-tuning using reflection-guided reward correction, showing progression from failure to success across four tasks: (a) carrot placement, (b) spoon placement, (c) eggplant basket placement, (d) cube stacking, (e) coke can placement, and (f) sprite can placement.

align with high-compactness regions, indicating its ability to perform more focused and discriminative frame selection throughout the iterations.

### D.2 Frame Index Trends Across Iterations

Figure 17 presents the iterative frame reduction behavior of ECRS across multiple videos, showing trends in average self-compactness (blue), GT-compactness (orange), and the number of selected frames (green). Self-compactness refers to the average index distance among the selected frame indices at each iteration, indicating how temporally clustered the selections are. GT-compactness measures the average index distance between selected frames and ground truth-relevant frames, reflecting how well the selected subset aligns with semantically important moments in the video. As iterations progress, both compactness metrics consistently decrease, demonstrating that ECRS not only reduces the number of selected frames but also improves their temporal tightness and alignment with ground truth. The number of selected frames drops sharply in early iterations and then stabilizes. This trend highlights ECRS's ability to efficiently eliminate redundant frames while preserving a compact, semantically meaningful subset aligned with human-annotated content.

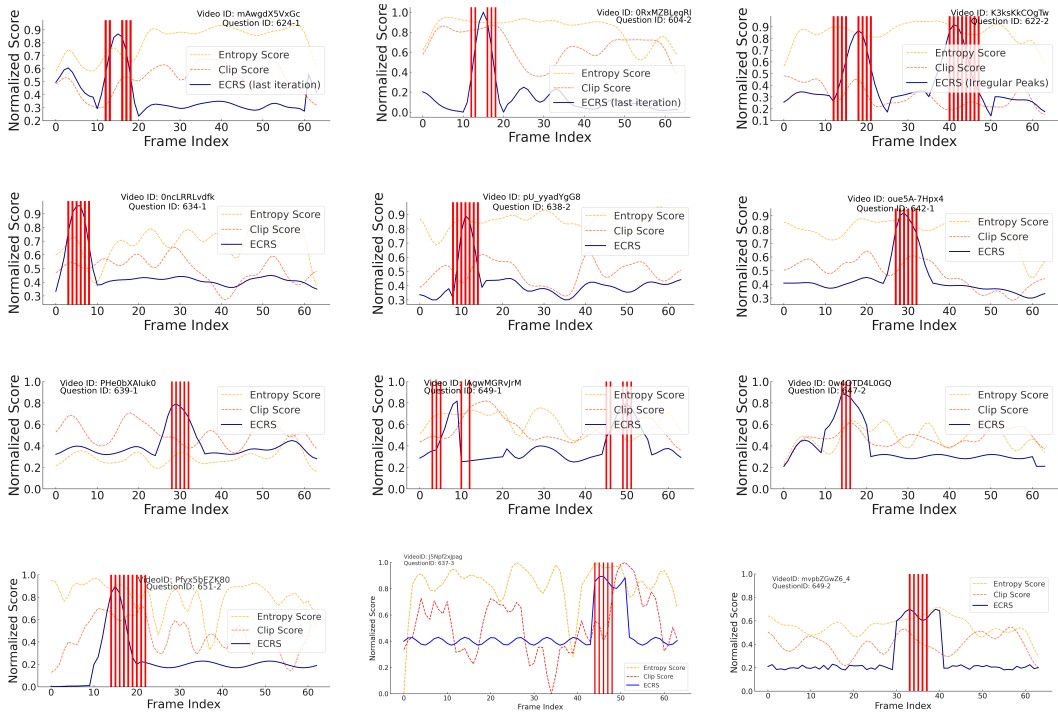

Figure 16: Comparison of frame selection strategies across multiple videos, illustrating how ECRS (final iteration), Entropy Score, and CLIP Score vary with frame index and influence selected keyframes (highlighted in red), along with their overlap with ground truth frames manually annotated by experts—refer to the VideoMME dataset for corresponding video IDs and validation details.

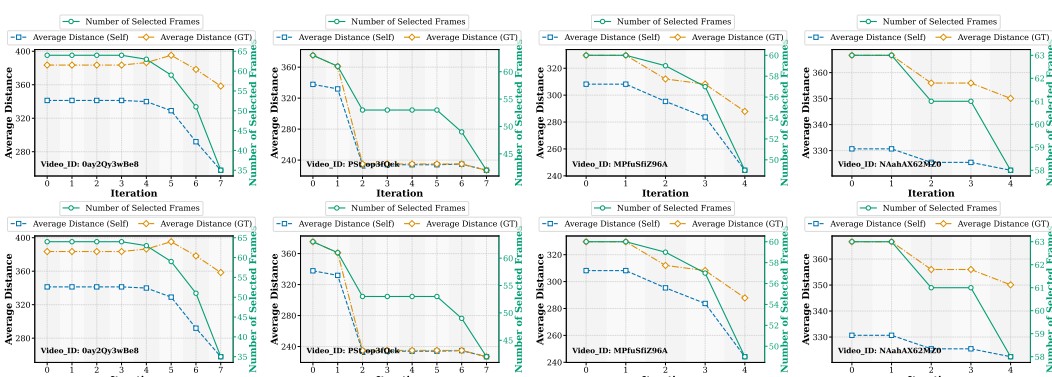

Figure 17: This figure shows tracks two compactness metrics - self-compactness (left y-axis, mean squared distance to selected frames' centroid) and GT-compactness (left y-axis;, mean squared distance to selected and ground truth centroids) and the number of selected frames) - along with frame count (right y-axis) across iterations (x-axis) for videos from the (Video-MME/MLVUI datasets. L), where lower left-axis values indicate tighter clustering.

## D.3 Comparison of Frame Selection Methods Across Varying Input Frame Numbers

Figure 18 shows that ECRS consistently outperforms other frame selection strategies across all frame counts on the VideoMME dataset. As the number of input frames increases from 8 to 64, accuracy improves across all methods, and the performance gap between strategies remains consistent. While these trends are observed across different base models, the results shown here use LLaVA-Video-7B, which exhibits particularly strong performance, further highlighting the effectiveness of ECRS in enhancing model accuracy through more informative frame selection.

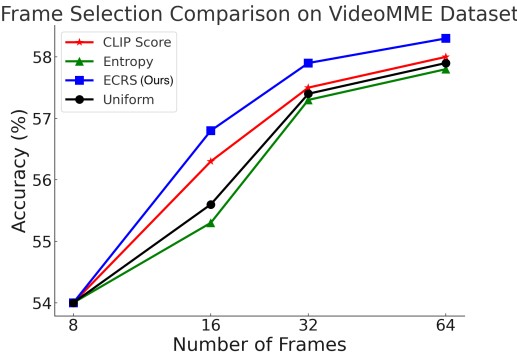

Figure 18: Frame selection strategies on VideoMME (LLaVA-Video-7B).

Table 5: Ablation study on the effect of Visual Tools across different base models and benchmarks.

| Model | Visual Tools | LongBench | EgoSchema | VideoMME |
|---|---|---|---|---|
| LLaVA-Video-7B | ✓ | **53.1** | **60.8** | **57.9** |
|  | ✗ | 46.1 | 56.7 | 56.3 |
| LLaVA-Video-72B | ✓ | **65.3** | **74.0** | **75.5** |
|  | ✗ | 60.9 | 75.0 | 67.3 |
| Qwen2-VL-7B | ✓ | **46.4** | **56.4** | **58.3** |
|  | ✗ | 44.9 | 55.6 | 53.8 |
| Qwen2.5-VL-72B | ✓ | **66.4** | **76.2** | **75.1** |
|  | ✗ | 62.6 | 75.7 | 74.3 |

### D.4  Ablation of Visual Tools

Table 5 presents an ablation study on the impact of incorporating Visual Tools across different base models and benchmarks. Across all settings, enabling visual tools consistently improves performance on LongBench, EgoSchema, and VideoMME. The gains are particularly notable for LLaVA-Video-7B and Qwen2-VL-7B, where enabling visual tools leads to substantial improvements—for instance, +7.0 on LongBench and +4.5 on VideoMME for LLaVA-Video-7B. While the performance gap is smaller for larger models like LLaVA-Video-72B and Qwen2.5-VL-72B, improvements are still evident on LongBench and VideoMME. These results indicate that visual tools provide valuable auxiliary signals that enhance multimodal reasoning, especially for smaller or less capable base models.

## Limitation

Despite employing keyframe selection, handling long videos remains challenging due to information redundancy and complex semantic relationships, which hinder the model's ability to achieve comprehensive and accurate understanding and reasoning. Additionally, the reflection and evaluation mechanisms rely on heuristic-based rules or templates, lacking adaptive and end-to-end learning capabilities, which may limit the effectiveness of automatic error correction.

## Social impact

Our method enhances video understanding and reasoning, which can benefit education, assistive technologies, and human-computer interaction. However, potential risks such as privacy concerns and the propagation of dataset biases should be carefully considered during deployment.

