# OpenReview forum: "ReAgent-V: A Reward-Driven Multi-Agent Framework for Video Understanding"
_NeurIPS.cc/2025/Conference — NeurIPS 2025 poster_

### Official Review · Reviewer_tdov · 2025-06-24

**Clarity:** 2
**Significance:** 3
**Originality:** 3
**Rating:** 5
**Confidence:** 4

**Summary:**

The paper proposes a video understanding framework that iteratively alternates between a target agent generating the answer and a critic agent proposing feedback for refinement. The target agent is provided iterative feedback and adjusts its answer with a multi-perspective reflection strategy which aggregates altered versions of the original answer across varying intensities (conservative, neutral and aggressive), allowing the model to correct previous errors in reasoning. The paper also proposes an entropy-guided frame selection scheme, maximising the information present in frames that have high similarity to the original query.

**Questions:**

The paper is well written and verifies the effectiveness of its contributions through the ablations. However, some experimental validation and clarity is missing. To improve my score further, the authors should answer the following:

1. Given the difference in model capacity and parameter count is unaccounted for, could the authors provide a fairer comparison with a similar capacity model, or further justification for the timing results presented in Table 3?
2. Could the authors clarify whether the CLIP embedding extraction is performed online during inference or as an offline pre-processing step, as this impacts the interpretation of the reported timings?
3. Could the authors provide an analysis of the failure modes that most frequently trigger the reflection mechanism? Specifically, which of the reasoning aspects outlined in the evaluation criteria tend to fail most often, necessitating refinement?
4. Could the authors provide an analysis of the confidence distributions for each of the reflection types (conservative, neutral, and aggressive)? This would help clarify whether the prompting strategy is manipulating the agent's output confidence.

**Ethical Concerns:**

["NO or VERY MINOR ethics concerns only"]

**Final Justification:**

Thank you to the authors for their engagement during the discussions. Following this, the ambiguities were resolved and some valuable insights were offered to better understand the paper. Overall, I am happy to accept.

**Limitations:**

Yes - Added in the supplementary

**Quality:**

3

**Strengths And Weaknesses:**

**Strengths:**
- The paper supports its two main contributions: the reflection mechanism and an entropy-guided frame selection scheme, with the ablation studies showing improvement to the model's overall performance when both elements are added.
- The core idea of employing iterative feedback between a target and critic agent is an interesting approach to addressing reasoning failures in video understanding. Furthermore, the proposed entropy-based is a strong idea for mitigating frame redundancy and increasing efficiency.
- Beyond its primary application, the framework demonstrates broader utility by showing it can be used to identify challenging samples for fine-tuning. The framework enhances sample efficiency and boosts performance when fine-tuning for the video reasoning/general benchmarks.

**Weaknesses:**
- The validity of the timing comparisons presented in Table 3 is questionable. The proposed method is compared against baselines utilising GPT-4, but the analysis does not account for the differences in model capacity and parameter count.
- The paper does not specify whether the extraction of CLIP embeddings is performed online during inference or offline as a pre-processing step. This also affects the reported timings in Table 3.
- The paper introduces a multi-perspective reflection strategy but does not provide an analysis of the conditions under which this reflection is most often required. While the supplementary material indicates the evaluation criteria in the prompt, it remains unclear which aspects of reasoning most frequently fail, thereby triggering the refinement process.
- An analysis of the confidence levels associated with each reflection type (conservative, neutral, aggressive) is notably absent. Without an examination of their respective confidence distributions, it is difficult to determine if the prompting strategy alters the output confidence. For instance, specifying an agent to be aggressive could cause over-confidence in its predictions and conversely under-confidence with the conservative agent.

**Minor Weaknesses:**
- Typo on Line 200: Should be a space between ReAgent-V and as
- Line 272 appears to have an incorrect reference to Figure 4. Likely should be Figure 5.
- The question "What is the telling..." in Figure 5 does not make sense and is inconsistent with the text on Line 287

---

> ### Author Rebuttal · Authors · 2025-07-31
>
> We are grateful for your comments and will address your concerns point by point in the following:
>
> **Q1:** Given the difference in model capacity and parameter count is unaccounted for, could the authors provide a fairer comparison with a similar capacity model, or further justification for the timing results presented in Table 3?
>
> **A1:** Thank you for the question. All inference time comparisons in Table 3 are conducted under the same model architecture and parameter scale. Taking LLaVA-Video-7B as an example, we control only whether the frame selection strategy is applied, in order to evaluate its impact on inference efficiency and accuracy. The results show that introducing the frame selection strategy leads to improvements in both efficiency and accuracy compared to the original model, demonstrating that the frame selection component in the ReAgent-V framework contributes positively to overall performance and ensures the objectivity and validity of the evaluation results.
>
> ---
>
> **Q2:** Could the authors clarify whether the CLIP embedding extraction is performed online during inference or as an offline pre-processing step, as this impacts the interpretation of the reported timings?
>
> **A2:** Thank you for the question, and we apologize for not making this point clear in the original manuscript. All inference-time measurements reported in Table 3 include the CLIP feature extraction stage, which is performed online during inference rather than as an offline pre-processing step.
>
> ---
>
> **Q3:** Could the authors provide an analysis of the failure modes that most frequently trigger the reflection mechanism? Specifically, which of the reasoning aspects outlined in the evaluation criteria tend to fail most often, necessitating refinement?
>
> **A3:**
> We sincerely thank you for your thoughtful question. To better understand which reasoning dimensions most frequently trigger the reflection mechanism, we analyzed the reflection rates on the VideoMME benchmark using ReAgent-V with Qwen-2.5-VL 7B as the base model.
> Our analysis shows that when categorizing videos by length - Short, Medium, and Long - the corresponding reflection rates are 36%, 45%, and 67%, respectively. This suggests a positive correlation between reflection frequency and task difficulty, as current models tend to perform worse on longer videos.
> In addition, we found that tasks related to Spatial Perception and Temporal Perception exhibit particularly high reflection rates, at 57% and 62% respectively. This indicates that current models are relatively weak in handling spatial and temporal reasoning in videos, often requiring multiple rounds of reflection to generate more accurate responses.
>
> ---
>
> **Q4:** Could the authors provide an analysis of the confidence distributions for each of the reflection types (conservative, neutral, and aggressive)? This would help clarify whether the prompting strategy is manipulating the agent's output confidence.
>
> **A4:**  Thank you again for your insightful question. We provide additional quantitative evidence based on experiments using ReAgent-V with Qwen-2.5-VL 7B as the base model on the VideoMME benchmark. We focused on the task with the highest reflection rate - Temporal Perception - and analyzed the average output confidence under the three reflection strategies. The conservative strategy produced an average confidence of 0.63, the neutral strategy 0.71, and the aggressive strategy the lowest at 0.45, with larger variance.
>
> However, as shown in Figure 4, all three strategies are able to correct a portion of the answers after reflection. Moreover, combining all three strategies yields a higher overall accuracy than using any single strategy alone. This suggests that even though the conservative and aggressive strategies tend to generate lower confidence under standard model decoding, they may still introduce correct information in certain cases, ultimately leading to improved answer accuracy.
>
> ---
>
> **Q5:**
> Typo on Line 200: Should be a space between ReAgent-V and as
> Line 272 appears to have an incorrect reference to Figure 4. Likely should be Figure 5.
> The question "What is the telling..." in Figure 5 does not make sense and is inconsistent with the text on Line 287
>
> **A5:**
> Really thank you for your careful reading - we will correct the missing space on Line 200 (“ReAgent-V as”), fix the incorrect figure reference on Line 272 (should be Figure 5), and revise the unclear phrasing in Figure 5 to align with the question on Line 287 in the final version. We will also carefully review the manuscript to ensure overall consistency and clarity in the final version.

---

> > ### Comment · Reviewer_tdov · 2025-08-01
> >
> > I thank the authors for their detailed rebuttal. The responses and the additional experiments have addressed my primary concerns.
> >
> > - **Regarding Q1 and Q2 (Timing and Implementation Details):** The clarification that the timing comparisons in Table 3 are conducted within the same model architecture (i.e. ablating the frame selection component) resolves my concern about an unfair comparison. Thank you also for confirming that the CLIP feature extraction is performed online and included in the reported timings. This information is important for a correct interpretation of the results, and I recommend adding this clarification to the final manuscript.
> > - **Regarding Q3 (Failure Mode Analysis):** The new analysis of reflection rates on the VideoMME benchmark is a strong addition. The finding that reflection is more frequent in longer videos and is particularly prevalent for tasks involving Spatial Perception and Temporal Perception provides valuable insight into the specific reasoning challenges that the proposed framework helps to mitigate. I recommend that this analysis be included in the revised paper, as it improves the justification for the reflection mechanism.
> > - **Regarding Q4 (Confidence Distribution Analysis):** I appreciate the authors providing the new experimental results on confidence distributions. The analysis is insightful, though surprising given the counter-intuitive finding that the 'aggressive' strategy yields lower average confidence. As a point of interest, could the authors offer a brief comment as to why the aggressive prompt might lead to lower output confidence? One might intuitively expect the opposite. A sentence or two speculating on this could add a further layer of depth.
> > - **Regarding Q5 (Minor Issues):** Thank you for agreeing to correct the these errors.
> >
> > In summary, the authors have addressed the weaknesses raised in my initial review. The addition of the new analyses on failure modes and confidence distributions will make this a much stronger and more complete paper. Additionally addressing my further question on Q4 will help convince me further.

---

> > > ### Author Response · Authors · 2025-08-04
> > >
> > > Response to Q1 & Q2 & Q3 & Q5:
> > > Thank you for your acknowlegement!
> > >
> > > Response to Q4 (Confidence Distribution)
> > >
> > > Really thank you for this interesting and insightful observation. We are more than happy to address this concern:
> > >
> > > The lower average confidence under the aggressive strategy may stem from two factors:
> > >
> > > (1) It involves more extensive edits, including changes to both reasoning and entity references, which naturally introduces more uncertainty;
> > >
> > > (2) In some cases, it may over-correct partially accurate answers, leading the model to be less confident even when the output improves.
> > >
> > > Importantly, as shown in Figure 4, this strategy still leads to better overall accuracy, suggesting that confidence scores are not always reliable indicators of correctness. We will incorporate this explanation into the final paper.

---

> > > > ### Comment · Reviewer_tdov · 2025-08-05
> > > >
> > > > Thank you to the authors for their reply. It is interesting to note that the model's confidence is potentially a reflection of the amount of edits it has made, rather than the strategy provided by the prompt. Overall, I am happy with the provided answers and have no further questions at this time. I will update my score pending the reviewer discussions that follow.

---

> > > > > ### Author Response · Authors · 2025-08-05
> > > > > **Official Comment by Authors**
> > > > >
> > > > > Dear Reviewer tdov,
> > > > >
> > > > > Thank you for your thoughtful follow-up and for engaging with our response. We're glad our clarification addressed your concerns, and we truly appreciate your time and feedback.

---

### Official Review · Reviewer_KqtJ · 2025-07-01

**Clarity:** 2
**Significance:** 2
**Originality:** 3
**Rating:** 4
**Confidence:** 3

**Summary:**

In the paper 'ReAgent-V: A Reward-Driven Multi-Agent Framework for Video Understanding' the authors tackle the problem of video understanding by proposing an agentic framework, based on reward generation via a reflection mechanism, that can also be used to filter high-quality frames for later fine-tuning, supervised or via reinforcement learning techniques.
Differently than current state-of-the-art methods, which use either high-quality annotations (expensive) or pretrained reward models (no real-time reasoning), the proposed algorithm aims to unlock iterative reasoning, self-correction and adaptability in more complex scenarios, not handled well by static approaches.
They test their framework on three tasks (visual understanding, visual reasoning and VLA alignment, 12 datasets in total) showing improved results over other baselines.

**Questions:**

How does frame selection reduce the inference time so much? (Table 3). Is the time required by this procedure offset by the reduction in computation?

Why is the amount of data used in Table 2 different for the various models? To validate the implicit claim (that ReAgent-V is better with less data) I'd like more information about how the other models behave with same amount of data, because it might be possible that the results would be similar with the same amount of data, or that ReAgent-V stays constant even after adding more data.

**Ethical Concerns:**

["NO or VERY MINOR ethics concerns only"]

**Final Justification:**

My main complaints were addressed and hence I believes it warrants a stronger score.

**Limitations:**

Not discussed in the main paper

**Quality:**

3

**Strengths And Weaknesses:**

Strengths

(Quality)
		The evaluation is extensive, using many different benchmarks. In general, the results are positive (but see weaknesses).
		The evaluation is carried out on more than one tasks, including a control-based one.

(Significance)
I like the entropy-calibrated frame selection and I think it is an interesting approach with insights for the community.

(Clarity)
		In general the paper reads well and the method is explained well


Weaknesses

(Quality and significance) Although the evaluation is extensive, on many different benchmarks, I feel the results are somewhat of limited impact, especially in video understanding. This problem is worsened by not having any discussion on the statistical significance of the results (even though the authors claim, in the checklist, to provide one) and by limited insights provided (see later). Moreover, the experimental section is primarily focused on testing VLMs with and without ReAgent (for example, Table 1). This limits understanding of the significance of the method: how does this proposed framework improve over the stated limitations of other SOTA methods (static approaches, or approaches using RL etc.)? To experimentally evaluate this, more comparisons with methods employing such approaches should be provided (for example, taken from the related works, [13][29] etc ) instead of testing multiple VLMs.

(Clarity)
Discussion/tests about the sensitivity of the entropy-based frame selection to the various hyperparameters is missing. There are some other problems:

i) the section 2.2 on tool selection has little detail on how it actually works concretely. For example: 'the target agent proactively selects a subset of tools T ′ ⊆ T based on its reasoning needs'. How?

ii) Similarly, 2.3 could be improved. I understand the specific details are in the appendix, but I feel the main text should still include more details.

iii)Small typo in line 200: missing space 'ReAgent-Vas...'

iv) In figure 5, I feel the question at the top has some errors: "What is the telling when the burger ..."

(Originality)
Besides the entropy-based frame selection, the proposed algorithm provides limited insights. When proposing a modular architecture built on top other more fundamental blocks (like VLMs), I think it is important to explain in detail why the chosen 'blocks' were picked and compare this with the decisions made by other state-of-the-art methods, outlining in particular how the new ideas mitigate the weaknesses of other methods, especially those mentioned in the introduction/abstract. Why, for example, you use those three specific strategies in the reflection phase? Do other state-of-the-art methods do something similar? Were there weaknesses in other SOTA models that are mitigated by using these three particular strategies that cannot be tackled, for example, with an alternative revision strategy? The ablation of Figure 4 is not enough to answer this. Without comparisons or an in-depth discussion, as a researcher I get limited insights for possible future works, only the 'quantitative' results, and the choices made seem pretty arbitrary. I'd like to see more discussions in this sense.

---

> ### Author Rebuttal · Authors · 2025-07-31
>
> Thank you for your detailed and critical feedback - we have carefully reviewed each point and respectfully address them below:
>
> **Q1:** (Quality and significance) Although the evaluation is extensive, on many different benchmarks, I feel the results are somewhat of limited impact, … To experimentally evaluate this, more comparisons with methods employing such approaches should be provided (for example, taken from the related works, [1][2] etc ) instead of testing multiple VLMs.
>
> References:
>
> [1]  Videorag: Retrieval augmented generation over video corpus.
>
> [2]  Reflexion: Language agents with verbal reinforcement learning.
>
> **A1:** Thank you for the insightful question. Our evaluation focuses on whether large vision-language models (LVLMs) benefit from the ReAgent-V framework under different parameter settings. As shown in the results, ReAgent-V consistently outperforms its base models and surpasses most strong baselines, such as InternVL-2.5-8B and BIMBA-LLaVA.  We further conducted additional experiments comparing ReAgent-V with the retrieval-based VideoRAG approach and a Reflexion-inspired variant, as shown in the tables below. Although Reflexion [2] was originally designed for language tasks, we adapted its core mechanisms. All models were evaluated under identical settings for fairness. Due to space constraints, detailed implementation specifics will be included in the final version release.
>
> The results show that ReAgent-V consistently outperforms [1], [2], and other baselines, demonstrating its effectiveness. All experiments were repeated three times, and our method achieves statistically significant improvements over models without ReAgent-V across all benchmarks (p < 0.01, two-tailed test), further validating its robustness.
>
> | Model (Backbone) | VideoMME (avg) | EgoSchema | LongBench |
> |------------------------|--------------|--------------|--------------|
> | ReAgent-V (Qwen2.5-VL-7B) | **60.7** | **61.9** | **54.3** |
> | VideoRAG (Qwen2.5-VL-7B) | 60.3 | 61.2 | 53.7 |
> | ReAgent-V (LLaVA-Video-7B) | **57.9** | **60.8** | **53.1** |
> | VideoRAG (LLaVA-Video-7B) | 57.2 | 60.3 | 52.6 |
>
> | Model (Backbone) | VideoMME (avg) | EgoSchema | LongBench |
> |------------------------|--------------|--------------|--------------|
> | ReAgent-V (Qwen2.5-VL-7B) | **60.7** | **61.9** | **54.3** |
> | Reflexion (Qwen2.5-VL-7B) | 50.6 | 53.2 | 43.8 |
> | ReAgent-V (LLaVA-Video-7B) | **57.9** | **60.8** | **53.1** |
> | Reflexion (LLaVA-Video-7B) | 46.7 | 50.5 | 42.8 |
>
>
> Beyond video understanding, we evaluate ReAgent-V on two additional tasks: Video LLM Reasoning and VLA Alignment. In Video LLM Reasoning, ReAgent-V outperforms the GRPO-trained model from [3] using only 45% of the training data. In VLA Alignment, our reflection-based reward replaces the hand-crafted cost function in GRAPE [4], yielding improved performance.
>
> References:
>
> [3] Video-r1: Reinforcing video reasoning in mllms, 2025.
>
> [4] GRAPE: Generalizing robot policy via preference alignment, 2025.
>
> ---
>
> **Q2:** (Clarity) Discussion/tests about the sensitivity of the entropy-based frame selection to the various hyperparameters is missing.
>
> **A2:** Thank you for the question. We provide a robustness analysis of the two key hyperparameters in Eq(4) - the scaling factor $k$ and threshold $\tau$ - using Qwen2-VL-7B on the short split of VideoMME.
>
> First, we fix $\tau = 0.7$ and vary $k \in \{1, 2, 3, 4\}$:
>
> **Table R1: Accuracy under different values of \(k\)**
> | \(k\) | Accuracy (%) |
> |------|--------------|
> | 1    | 73.5         |
> | 2    | 71.5         |
> | 3    | 74.5         |
> | 4    | 72.8         |
>
> Next, we fix $k = 1$ and vary $\tau \in \{0.5, 0.6, 0.7, 0.8\}$:
>
> **Table R2: Accuracy under different values of $\tau$**
>
> | $\tau$ Value | Accuracy (%) |
> |---------|--------------|
> | 0.5     | 71.8         |
> | 0.6     | 72.3         |
> | 0.7     | 73.5         |
> | 0.8     | 71.2         |
>
> The results show that our method is stable across a wide range of parameter values. Overly strict thresholds (e.g., $\tau = 0.8$) may select too few frames and hurt performance. We adopt $k = 1$ and $\tau = 0.7$ as default for a good balance of robustness and effectiveness, and will include this in the revised paper.
>
> ---
>
> **Q3:** There are some other problems:
>
> i) the section 2.2 on tool selection has little … How?
>
> ii) Similarly, 2.3 could be improved. I understand the specific details are in the appendix, but I feel the main text should still include more details.
>
> iii)Small typo in line 200: missing space 'ReAgent-V as...'
>
> iv) In figure 5, I feel the question at the top …burger ..."
>
> **A3:**  Thank you for your careful reading and valuable comments. We will fix the typo on line 200 (“ReAgent-Vas...”) and revise unclear phrasing in Figure 5 (e.g., “What is the telling…”). We also acknowledge that Sections 2.2 and 2.3 lack implementation details. For tool selection, the base model determines which tools to invoke based on the question, the video content, and its own reasoning needs. In the revision, we will clarify how base models select tools based on the question and video content, and provide a more detailed explanation of tool selection and reflection strategies in the main text rather than in the appendix
>
> ---
>
> **Q4:** (Originality) Besides the entropy-based frame selection, the proposed algorithm provides limited insights. … I get limited insights for possible future works, only the 'quantitative' results, and the choices made seem pretty arbitrary. I'd like to see more discussions in this sense.
>
> **A4:** Thank you again for the insightful comments. The three reflection strategies are designed to mitigate overconfidence in single-strategy responses, which often cause hallucinations and errors [1, 2]. Each strategy provides a distinct perspective, and their combination enables the model to integrate complementary signals, leading to more accurate answers. As shown in Figure 4, this combined approach outperforms individual strategies on benchmarks such as VideoMME, LongBench, and EgoSchema.
>
> To further support this, we analyzed the VideoMME benchmark using ReAgent-V with Qwen2.5-VL-7B, focusing on the Temporal Perception task where reflection is frequently triggered. The average output confidences for the conservative, neutral, and aggressive strategies were 0.63, 0.71, and 0.45, respectively, with the aggressive strategy showing the highest variance. This suggests base models are less likely to produce conservative or aggressive responses due to lower confidence, but by explicitly prompting these strategies and merging their outputs, we can capture cases that single-strategy responses may overlook. ReAgent-V’s reward module not only generates reward reports to guide data selection for GRPO/DPO training but also directly improves base model reasoning through answer revision - a capability absent in prior agent frameworks. In the Video LLM Reasoning task, using ReAgent-V’s reward reports to filter GRPO training data yields more efficient performance gains than standard methods.
>
> References:
>
>  [1] Analyzing and mitigating object hallucination in large vision-language models
>
>  [2] VL-Uncertainty: Detecting hallucination in large vision-language model via uncertainty estimation
>
> ---
>
> **Q5:** How does frame selection reduce the inference time so much? (Table 3). Is the time required by this procedure offset by the reduction in computation?
>
> **A5:** Thank you for the question. While ECRS-based frame selection introduces some overhead, it significantly reduces the number of frames passed to the vision-language model - the main inference bottleneck - resulting in efficiency gains that far outweigh the selection cost.
>
> ---
>
> **Q6:** Why is the amount of data used in Table 2 different for the various models? To validate the implicit claim (that ReAgent-V is better with less data) … or that ReAgent-V stays constant even after adding more data.
>
> **A6:** Thank you for this interesting and careful question. In the Video LLM Reasoning application, ReAgent-V is used to filter GRPO training data via a reflection-based mechanism. As pointed out in [1], more challenging examples often lead to greater improvements in GRPO training.
>
> Using the dataset from [2], we leveraged ReAgent-V’s reflection signals to identify difficult samples - those that triggered reflection - and retained only these difficult samples, comprising 45% of the original data. Training the base model from [2] on this subset under the same GRPO setup resulted in better performance than using the full dataset. This shows that ReAgent-V not only improves reasoning but also serves as an effective data filtering mechanism for reinforcement learning pipelines.
>
> To further validate this, we conducted a control experiment by randomly selecting an equal-sized subset (52k samples) from the original training set. As shown below, the model trained on randomly selected data underperformed compared to both the full-data baseline and our reflection-based filtering:
>
> | Method               | Steps | #Data | VSI-Bench | VideoMMMU | MMVU | MVBench | TempCompass | VideoMME |
> |---------------------|--------|--------|------------|-------------|-------|-----------|----------------|-------------|
> | Vanilla GRPO [2]    | 16     | 116k   | 32.3       | 45.8        | 60.6  | 60.9      | 69.8           | 53.8        |
> | GRPO + Random select  | 16     | 52k    | 31.2       | 44.3        | 59.1  | 59.6      | 68.4           | 52.5        |
> | GRPO + ReAgent-V    | 16     | 52k    | **33.1**   | **47.9**    | **63.0** | **61.4**  | **70.3**       | **54.2**    |
>
> These results highlight the importance of selecting informative training samples rather than relying on quantity, and confirm the utility of ReAgent-V as a high-quality data selector for downstream learning tasks.
>
>
> References:
>
> [1] Sota with less: Mcts-guided sample selection for data-efficient visual reasoning self-improvement, 2025.
>
> [2] Video-r1: Reinforcing video reasoning in mllms, 2025.

---

> > ### Comment · Reviewer_KqtJ · 2025-08-03
> > **Response**
> >
> > I thank you for your thorough rebuttal and the new details. My main complaints are addressed and I will update my score.

---

> > > ### Author Response · Authors · 2025-08-05
> > > **Official Comment by Authors**
> > >
> > > Dear Reviewer KqtJ,
> > >
> > > We are glad that we have addressed all your concerns! We deeply appreciate your positive feedback and your decision to raise your score! We sincerely thank you for the time and effort you have dedicated to reviewing our paper and helping us further improve our work. We will carefully follow your suggestions and incorporate all updates into the new version. Thank you again for your valuable support!

---

### Official Review · Reviewer_dAzj · 2025-07-03

**Clarity:** 3
**Significance:** 3
**Originality:** 3
**Rating:** 4
**Confidence:** 3

**Summary:**

The paper proposes an agentic video understanding framework. It involves a frame selection to pick salient input frames. It also provides a tool factory to assist model for video understanding. To supervise the model, it designs a critic agent for evaluating model results and providing evaluation report for improvement guidance. The target agent then refine based on three strategies to produce better predictions.

**Questions:**

Please provide analysis for model robustness on hyper-parameters and discussion on the effect of stronger MLLM backbone.

**Ethical Concerns:**

["NO or VERY MINOR ethics concerns only"]

**Final Justification:**

I appreciate the rebuttal from the authors. After reading it and the opinions of the reviewers, I will maintain my positive rating.

**Quality:**

3

**Strengths And Weaknesses:**

Strengths:
- The paper proposes a new framework for agentic video understanding, and shows the framework is versatile and can support both test-time improvement and training time data construction.
- The proposed entropy-based frame selection is interesting and has demonstrated consistent improvement.
- The paper writing is easy to follow with clear structure.
- The experimental results are reported for three settings to verify method effectiveness.

Weakness:
- The selection threshold used in Eq(4) involves multiple parameters. It is unclear if the method is robust to those parameters or requires careful parameter tuning.
- In Table 1, the best performance is achieved with Qwen 2.5 72B, which is a larger MLLM than those used in [8]. Results with similar scale of MLLM shows mixed performance. It makes it hard to understand the improvement from the MLLM vs improvement from the method.

---

> ### Author Rebuttal · Authors · 2025-07-31
>
> Thank you for your thoughtful and constructive comments, we address each of your concerns in detail below:
>
> **Q1:** The selection threshold used in Eq(4) involves multiple parameters. It is unclear if the method is robust to those parameters or requires careful parameter tuning.
>
> **A1:** We sincerely thank you for this thoughtful question! This is indeed an important aspect that we should have clarified in the main text. Due to time limitations, we only included partial results earlier. Here, we conduct a more comprehensive robustness study on the parameters used in Eq (4), specifically the scaling factor \(k\) and the threshold $\tau$, using Qwen2-VL-7B as the base model on the short split of the VideoMME. We vary the scaling parameter $k \in \{1, 2, 3, 4\}$ while keeping $\tau = 0.7$, and observe the following results:
>
> **Table R1:** Robustness Analysis of Parameters in Eq(4) on VideoMME (Short Split, Qwen2-VL-7B)
> | \(k\) Value | Accuracy (%) |
> |------------|--------------|
> | 1          | 73.5         |
> | 2          | 71.5         |
> | 3          | 74.5         |
> | 4          | 72.8         |
>
> We also fix $k = 1$ and vary the threshold $\tau \in \{0.5, 0.6, 0.7, 0.8\}$:
> | ($\tau$) Value | Accuracy (%) |
> |---------------|--------------|
> | 0.5           | 71.8         |
> | 0.6           | 72.3         |
> | 0.7           | 73.5         |
> | 0.8           | 71.2         |
>
> Overall, the results show that our method is robust to the choice of parameters. Performance remains relatively stable across a wide range of $k$ and $\tau$ values. We also note that setting $\tau$ above 0.8 may result in failure to select enough frames (e.g., 32), as the threshold becomes overly strict. Based on these observations, we adopt default values of $k = 1$ and $\tau = 0.7$, which provide strong and stable performance without requiring fine-grained tuning.
>
> ---
>
> **Q2:** In Table 1, the best performance is achieved with Qwen 2.5 72B, which is a larger MLLM compared to those used in prior work. Results with models of similar scale show mixed performance, making it difficult to disentangle the improvement brought by the MLLM itself from the gains introduced by the proposed method. (Please provide analysis for model robustness on hyper-parameters and discussion on the effect of a stronger MLLM backbone.)
>
>
> **A2:** Thank you once again for your insightful question. To clarify the source of performance improvements, we conducted controlled experiments in Table 1 across different models (LLaVA-Video and Qwen2.5-VL) at both 7B and 72B scales. In both settings, incorporating ReAgent-V consistently leads to significant improvements over the corresponding base models. This indicates that the gains are not merely due to the use of a stronger backbone, but rather stem from our proposed reflection mechanism and reward-driven reasoning framework. These components enable the ReAgent-V framework to exhibit strong robustness across different model capacities.

---

### Official Review · Reviewer_Nwjt · 2025-07-05

**Clarity:** 3
**Significance:** 3
**Originality:** 2
**Rating:** 4
**Confidence:** 4

**Summary:**

This paper proposes ReAgent-V, a reward-driven multi-agent framework for video understanding. The framework consists of three parts. First, it selects frames relevant to the query by an entropy-calibrated selection strategy. Second, it invokes various tools (e.g., text extraction, object detection) to assist the reasoning process. Finally, it introduces a multi-perspective reflection mechanism, where a critic agent produce suggestions to guide the target agent to refine the answer. The paper compare ReAgent-V to multiple video understanding baselines on multiple video tasks. The paper also quantitatively and qualitatively analyzes the contributions of each component

**Questions:**

1.Could the authors provide more quantitative results on the selection quality of ECRS? Are there failure modes where ECRS excludes or over-selects critical frames?

2.Could the authors clarify how the conservative, neutral, and aggressive reflection strategies differ in practice by cases? How robust are these strategies to prompt during reflection?

**Ethical Concerns:**

["NO or VERY MINOR ethics concerns only"]

**Limitations:**

Yes

**Quality:**

3

**Strengths And Weaknesses:**

Strengths
1.The proposed framework addresses the important limitation of static, single-pass video reasoning in large vision-language models. By integrating real-time reward generation and multi-perspective reflection, ReAgent-V enables dynamic, self-correcting, and tool-augmented inference, which is highly relevant for complex video understanding scenarios.
2.The experiments are solid and comprehensive. The authors conduct extensive experiments on 12 datasets across three applications—video understanding, video LLM reasoning, and vision-language-action alignment. Results show clear performance gains over prior baselines. The paper also includes ablations for frame selection and reflection mechanisms, showing meaningful improvements in both accuracy and efficiency.
3.The agent frame work is extensible to different backbone models.

Weaknesses
1.The paper lacks quantitative analysis on the accuracy of ECRS. There’s no data showing how well frames selected by ECRS align with actual keyframes related to the query.
2.The paper uses prompts to control the behavior of the three different reflection agents (conservative, neutral, and aggressive), but it doesn’t provide quantitative evidence showing whether these agents truly exhibit distinct behavior patterns.

---

> ### Author Rebuttal · Authors · 2025-07-31
>
> We appreciate your insightful feedback, and we respond to your points individually as follows:
>
> **Q1:** The paper lacks quantitative analysis on the accuracy of ECRS. There’s no data showing how well frames selected by ECRS align with actual keyframes related to the query. Could the authors provide more quantitative results on the selection quality of ECRS?
>
> **A1:** Thank you for your insightful question! To quantitatively assess how well ECRS-selected frames align with ground-truth keyframes, we randomly sampled 100 videos from the VideoMME dataset, LongBench dataset and manually annotated the keyframes for each. We then compared ECRS against two common baselines - CLIP-based similarity and Entropy-based selection - on both VideoMME and LongBench benchmarks using five evaluation metrics: Recall, Precision, IoU, F1 Score, and Redundancy (defined as 1 minus the average pairwise CLIP similarity, where lower is better).
>
> As shown in Table R1, ECRS significantly outperforms both baselines across all metrics. On VideoMME, ECRS achieves markedly higher Recall (0.891), Precision (0.872), IoU (0.816), and F1 Score (0.623), while reducing Redundancy (0.182) compared to CLIP (0.763/0.649/0.607/0.486/0.242) and Entropy (0.604/0.570/0.455/0.421/0.279). On LongBench, which involves long-form videos with sparse but critical evidence, ECRS maintains top performance with Recall (0.852), Precision (0.841), and IoU (0.891), again outperforming CLIP and Entropy by large margins and demonstrating its robustness in both short and long video settings.
>
> ---
>
> **Table R1:** Quantitative Evaluation of Frame Selection Accuracy
> | **Dataset** | **Method** | **Recall** | **Precision** | **IoU** | **F1 Score** | **Redundancy** |
> | ------------- | ---------- | ----------- | -------------- | --------- | ------------- | --------------- |
> | **VideoMME** | **ECRS** | **0.891** | **0.872** | **0.816** | **0.623** | **0.182** |
> |  |CLIP | 0.763 | 0.649 | 0.607 | 0.486 | 0.242 |
> |  |Entropy | 0.604 | 0.570 | 0.455 | 0.421 | 0.279 |
> | **LongBench** | **ECRS** | **0.852** | **0.841** | **0.891** | **0.601** | **0.196** |
> |  |CLIP | 0.621 | 0.618 | 0.379 | 0.468 | 0.257 |
> |  |Entropy | 0.463 | 0.339 | 0.312 | 0.427 | 0.287 |
>
> ---
>
> **Q2:** Could the authors clarify how the conservative, neutral, and aggressive reflection strategies differ in practice by cases? How robust are these strategies to prompt during reflection?
>
> **A2:** We sincerely thank you again for this interesting question. The prompts for all three strategies are provided in Appendix B.2. Specifically, when the base model's answer is deemed unsatisfactory, the conservative strategy only modifies the final answer while leaving the rest of the response unchanged. The neutral strategy makes moderate adjustments, including correcting the answer and any relevant nouns. The aggressive strategy goes further by not only correcting the answer and related nouns but also modifying the logical relationships within the sentence. For an analysis of the robustness of each strategy, please refer to Figure 4 in the main text. We report both the probability of making edits and the post-editing accuracy for each strategy. The results show that while each individual strategy can improve the response to some extent, combining all three strategies yields the highest overall accuracy.

---

### Decision · Program_Chairs · 2025-09-17

**Decision:**

Accept (poster)

**Comment:**

This paper proposes to have a critic agent provides feedback to a target agent, which then refines its answer to video understanding problems using a multi-perspective reflection mechanism. The paper also introduces an entropy-calibrated frame selection method to improve performance and reduce computational cost.

All four reviewers found merit in the paper, appreciating the novelty of the agentic framework, the effectiveness of the proposed frame selection method, and the extensive experimental validation. Some of the initial concerns such as missing ablations and comparisons, clarity, and originality were largely addressed during the rebuttal phase, through additional experiments such as quantitative validation of ECRS, hyperparameter sensitivity analysis, and new comparisons against baselines. The paper overall makes valuable contributions to the venue.